

# High spatial resolution mapping of aerosol composition and sources in Oakland, California using mobile aerosol mass spectrometry

Rishabh U. Shah[1,2], Ellis S. Robinson[1,2], Peishi Gu[1,2], Allen L. Robinson[1,2], Joshua S. Apte[3], and Albert A. Presto[1,2]

[1]Mechanical engineering, Carnegie Mellon University, Pittsburgh PA
[2]Center for Atmospheric Particle Studies, Carnegie Mellon University, Pittsburgh PA
[3]Department of Civil, Architectural and Environmental Engineering, University of Texas at Austin, Austin TX

**Correspondence:** Albert A. Presto (apresto@andrew.cmu.edu)

**Abstract.** We investigated spatial and temporal patterns in concentration and composition of sub-micron particulate matter ($PM_1$) in Oakland, California in the summer of 2017 using an aerosol mass spectrometer mounted in a mobile laboratory. We performed $\sim 160$ hours of mobile sampling in the city over a 20-day period. Measurements are compared for three adjacent neighborhoods with distinct land uses: a central business district ("downtown"), a residential district ("West Oakland"), and

a major shipping port. The average organic aerosol (OA) concentration is 5.3 $\mu gm^{-3}$ and contributes $\sim 50\%$ of the $PM_1$ mass. OA concentrations in downtown are, on average, 1.5 $\mu gm^{-3}$ higher than in West Oakland and Port. We decomposed OA into three factors using positive matrix factorization: hydrocarbon-like OA (HOA; 20% average contribution), cooking OA (COA; 25%) and semi-volatile oxidized OA (SV-OOA; 55%). The collective 45% contribution from primary OA (HOA + COA) emphasizes the importance of primary emissions in Oakland. The dominant source of primary OA shifts from HOA-rich

in the morning to COA-rich after lunch time. COA in downtown is consistently higher than West Oakland and Port due to a large number of restaurants. HOA exhibits variability in space and time. Morning-time HOA concentration in downtown is twice that in Port, but Port HOA increases more than two-fold during mid-day, likely because trucking activity at the Port peaks at that time. Despite the expectation of being spatially uniform, SV-OOA also exhibits spatial differences. Morning-time SV-OOA in downtown is roughly 25% ($\sim 0.6$ $\mu gm^{-3}$) higher than the rest of Oakland. Even as the entire domain approaches

a more uniform photo-chemical state in the afternoon, downtown SV-OOA remains statistically higher than West Oakland and Port, suggesting that downtown is a microenvironment with higher photochemical activity. Higher concentrations of particulate sulfate (also of secondary origin) with no direct sources in Oakland further reflect higher photochemical activity in downtown. A combination of several factors (poor ventilation of air masses in street canyons, higher concentrations of precursor gases, higher concentrations of the hydroxyl radical) likely result in the proposed high photochemical activity in downtown. Lastly,

through Van Krevelen analysis of elemental ratios (H/C, O/C) of the OA, we show that OA in Oakland is more chemically reduced than several other urban areas. This underscores the importance of primary emissions in Oakland. We also show that mixing of oceanic air masses with these primary emissions in Oakland is an important processing mechanism that governs the overall OA composition in Oakland. The findings of this study are important because the pollutants we find contributing the most to OA variability, both of primary and secondary origin, are ubiquitous in other urban locations.

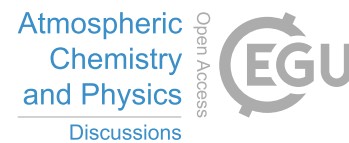

*Copyright statement.* TEXT

# 1 Introduction

Organic aerosol (OA) contributes a significant fraction of the total ambient particulate matter (PM) mass (Zhang et al., 2007), which is of utmost concern for its detrimental effects on human health (Apte et al., 2015) and the Earth's radiative budget
(Myhre et al., 2013). However, owing to the tens of thousands of different emitted organic species and their chemical and physical transformation in the atmosphere, the concentration and composition of OA remains complex to characterize (Hallquist et al., 2009; Jimenez et al., 2009; Tsigaridis et al., 2006; Goldstein and Galbally, 2007).

Concentrations of PM and other pollutants are spatially variable in urban areas and these spatial variations drive differences in human exposures. For example, concentrations of ultrafine particles, NO, CO, and particulate black carbon (BC) are enhanced
near highways by a factor of 2-3 relative to areas > 100m from roadways (Choi et al., 2012; Apte et al., 2017; Saha et al., 2018a). Spatial variations in PM mass are more modest (e.g., only a $\sim 25\%$ increase near roadways; Saha et al., 2018a), but are convolved with significant variations in composition (Mohr et al., 2015; Enroth et al., 2016; Canagaratna et al., 2010). In the near-source region fresh emissions of BC and primary OA rapidly mix with background air, reducing concentrations through both dilution and OA partitioning. This mixing occurs over the space of tens to hundreds of meters and is expected to require
timescales of minutes to hours (Robinson et al., 2010).

Studying these intra-urban PM variations can reveal the major sources influencing local air quality and help inform mitigation strategies. In particular, mobile sampling enables deployment of high-time-resolution measurements that can identify specific PM sources. For example, Li et al. (2018) showed that emissions of primary OA and BC drive much of the spatial variation in $PM_{2.5}$ observed in Pittsburgh. Apte et al. (2017) used mobile BC measurements to identify hotspots associated with vehicle
traffic and industrial activities. Some studies have deployed aerosol mass spectrometers (AMS) on mobile platforms (Mohr et al., 2015; Elser et al., 2016; Von Der Weiden-Reinmüller et al., 2014). Factor analysis of AMS data allows for identification of chemically-specific OA sources, such as traffic, restaurant cooking, and home heating. For instance, Elser et al. (2016) found enhancements of hydrocarbon-like OA (HOA, from traffic emissions) and BC on busy roads during times of peak traffic and that traffic emissions were the dominant contributor to the urban increments of $PM_{2.5}$. In addition to traffic emissions, Mohr
et al. (2015) found cooking and biomass burning emissions to be prominent contributors to primary OA in Barcelona.

Quantifying PM and OA spatial gradients in urban areas, and identifying the sources driving those gradients, is important because more than half of the world's population lives in urban areas (United Nations, 2014). Large populations may therefore live or work in areas with elevated emissions and/or high OA concentrations. Identifying such areas, and the sources driving elevated concentrations, has far-reaching implications for both reducing human PM exposures and addressing socio-economic
disparity in exposure to pollution on an intra-city scale.

This study presents results of mobile measurements conducted in Oakland, California. Oakland is a densely populated ($\sim$ 2900 inhabitants $km^{-2}$) pollution-source-rich city. It has a poverty rate (fraction of population below poverty line) of 18.9%, roughly twice as much as that of the collective San Francisco (SF) Bay area (9.2%) (US Census, 2016). It therefore offers





a good test case for investigating spatial variations in OA, how these variations are influenced by high industrial presence in residential areas, and how these variations overlay with population. Oakland has a unique land-use feature in that a 1 km$^2$ downtown, a 6 km$^2$ mixed industrial and residential district, and one of the largest US shipping ports (5 km$^2$) all lie within a short spatial transect of 4 km (Figure 1A).

Several prior studies have focused on air quality in Oakland because of the influence of the port and associated trucking activity through the residential district. Fisher et al. (2006) was one of the earlier studies addressing the heavy-duty diesel drayage trucks in Oakland as a mobile source. Since then, several legislations such as enforcement of diesel particulate filters in drayage truck exhausts (CARB, 2011), improved truck queuing system to reduce idling (Giuliano and O'Brien, 2007), usage of low-sulfur fuel in ships approaching the port (CARB, 2009), and keeping shipping logistics gates open in the evening

to dilute daytime congestion of drayage trucks (Port of Oakland, 2016) have been imposed to reduce port emissions and to improve Oakland's air quality. Consequently, recent studies have found substantial reductions in emissions from both drayage trucks and ships in Oakland (Preble et al., 2015; Dallmann et al., 2011; Tao et al., 2013). However, the presence of a large port and high drayage truck activity is still a large area source adjacent to the predominantly residential West Oakland district. Additionally, on the other side of this residential district lies downtown Oakland, which contains a mix of common urban

emission sources (e.g., vehicular traffic emissions). Four interstate highways (I-80 and its arteries I-580, I-880, I-980) closely flank the residential district such that the largest spatial lag from any point inside of West Oakland to the nearest highway is 1 km (Apte et al., 2017).

    The objective of this study is to determine which emission sources most strongly impact spatial patterns in the local air quality of Oakland. We use mobile sampling with aerosol mass spectrometry (AMS) to investigate spatial gradients in con-

centrations and chemical composition of OA across the three distinct areas of Oakland: port, residential West Oakland, and downtown. Additionally, using positive matrix factorization of AMS data (Ulbrich et al., 2009; Paatero and Tapper, 1994), we perform chemical source apportionment of the OA in Oakland. Results of this study not only provide valuable information on composition and source-assessment of PM$_1$ in Oakland, but source apportionment analysis shows how much the air quality in Oakland is impacted by local emissions versus chemically processed OA.

## 2 Experimental section

### 2.1 Mobile sampling

We conducted mobile sampling between 10$^{\text{th}}$ July and 2$^{\text{nd}}$ August 2017 in Oakland, CA using a mobile laboratory. Data were collected as part of the Center for Air, Climate, and Energy Solutions (CACES) air quality observatory (Zimmerman et al., 2018). The mobile laboratory is an instrumented Nissan 2500 cargo van, previously described by Li et al. (2016, 2018). Figure

1A shows a map of the sampling domain. We sampled on all streets in the domain that were open to public traffic.

    We divided the sampling domain into three main areas: Port, West Oakland and downtown. Owing to the relatively large size and road length density (16.6 km of road per km$^2$) of West Oakland, we further divided it into seven polygons. Port, while larger in area than West Oakland and downtown, has a very low road length density (2.6 km/km$^2$), because most of the area is



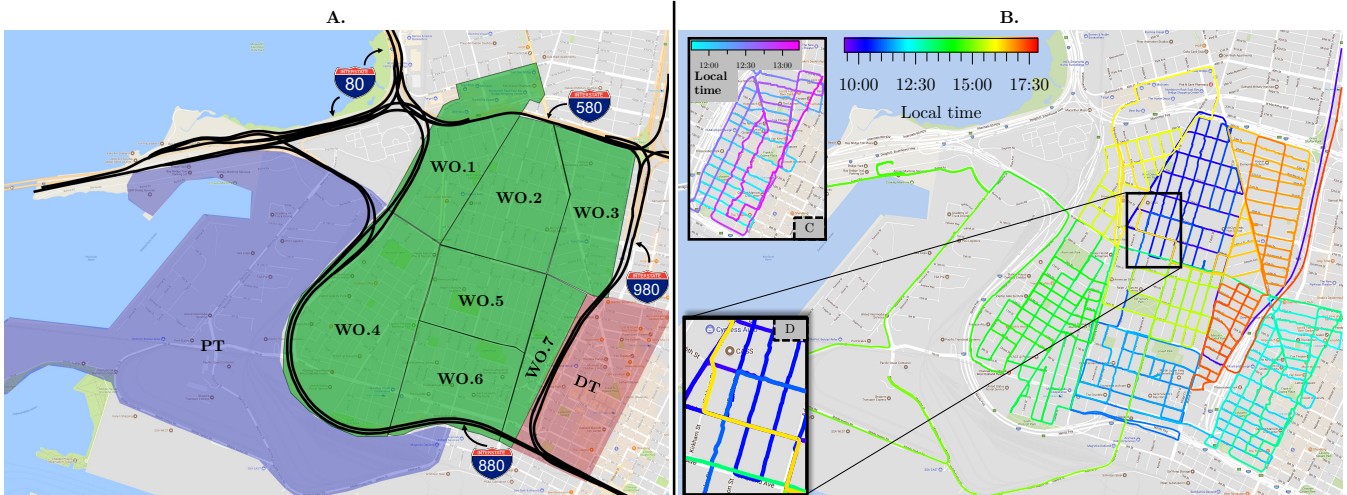

**Figure 1.** (a) Sampling domain with three major areas: PT (Port), WO (West Oakland) and DT (Downtown). The seven internal WO polygons are enumerated. (b) driving route on a typical day, showing shuffled order of polygon visits. (c) the typical driving route in downtown. (d) a zoom-in illustrating a typical repeat-sampling on the same day during inter-polygon transit

used for parking drayage trucks and storing shipping containers. Downtown has the highest road length density (24.8 km/km$^2$), as would be expected in a central business district. The predominant winds in the domain are from between the north-west and south-west, as shown in Figure S16 and discussed later in Section 3.3.

Figure 1B shows the driving route on a typical day of sampling, colored by the time of day. The order in which we visited 5 the nine polygons was shuffled daily to avoid systematically over- or under-sampling any polygon(s) in the morning, mid-day or late afternoon. Within a polygon, we employed a spiral driving pattern similar to that described by Apte et al. (2017). An exception was downtown, where, owing to alternating one-way streets, a zigzag driving pattern was used as shown in inset C in Figure 1.

## 2.2 Instrumentation

10 All instruments in the mobile laboratory were powered by a 110 V, 60 Hz alternator coupled to the van's engine. A 0.5" OD stainless steel tube carried the samples from the roof of the van ($\sim$ 3 m above ground level) to instruments as well as a mechanical backing pump. An in-line cyclone separator was installed upstream of instruments. Flow drawn by the backing pump was controlled by a needle valve such that the total flow drawn at the sampling inlet ($\sim$ 15 slpm) corresponded to a 2.5 $\mu$m cut-size diameter for the cyclone separator.

15 We used a high resolution time-of-flight aerosol mass spectrometer (HR-ToF-AMS, Aerodyne Research Inc.; Decarlo et al., 2006; Jayne et al., 2000) for measuring mass concentrations of non-refractory PM$_1$. The AMS was operated in V-mode with 20 s averaging of mass spectra. We did not collect particle time-of-flight data (used for size distribution measurements) because the additional averaging time required for collecting size distributions would decrease the overall sampling rate, compromising




the goal of collecting in-motion samples with high spatial resolution. Flow was dried to <5% relative humidity prior to the AMS using a Nafion drier (MD-110-24, PermaPure). A seven-wavelength, dual-spot aethalometer (AE33, Magee Scientific) measured concentrations of black carbon with 1 min averaging per sample. We also measured CO (T300U, Teledyne API), $CO_2$ (LI-820, LI-COR), and particle number concentration (200P, Aerosol Dynamics Inc.) at 1 Hz sampling rate. A GPS sensor

(BE-2200, Bad Elf) recorded GPS coordinates every second.

## 2.3  Data analysis

*Timestamps and GPS co-ordinates.* We first adjusted the recorded timestamps on all instrument samples based on the predetermined instrumental response times. Response times were measured by releasing a tracer at the sample inlet and recording the time lag in response from instruments while the van was stationary. This adjustment was done so as to assign each data point

to the time the sample entered the inlet (as opposed to the time the data point was recorded by the instrument). Additionally, a sampling duration offset was applied to the AMS data timestamps. This is because each AMS measurement is an average of mass spectra collected for 20 s and the timestamp is assigned at the end of the 20 s period. To ensure that the sample was spatially representative of the distance traveled by the van during the 20 s sampling interval, each AMS sample was advanced 10 s in time so as to assign the measured concentration to the middle of the 20 s sampling interval, instead of the end. Next,

upon alignment with GPS data, we assigned spatial coordinates to all instrument samples.

*Spatial analyses.* For spatial aggregation, we used a procedure similar to the "road-length snapping" procedure used by Apte et al. (2017). However, since AMS samples are recorded every 20 s and the van was driven at an average speed of 10 ms$^{-1}$, each AMS data point occurred roughly 200 m apart. We obtained a geospatial shapefile of Oakland's public streets from the Alameda County online archive (Alameda County, 2017). We created artificial points ("magnets") every

200 m along all streets. Multiple lanes of major surface streets were merged (i.e., opposing traffic lanes of large roads became single roadway centerlines) before creating magnets along them. This list of magnets was then matched against the GPS coordinates assigned to all AMS data points in MATLAB R2015a (MathWorks, Natick, MA) by calculating $d_{mag} = \sqrt{(y_{sample} - y_{magnet})^2 + (x_{sample} - x_{magnet})^2}$, where $y$ are latitudes and $x$ are longitudes. Each AMS sample was assigned to the magnet for which it had the smallest $d_{mag}$. A snapping threshold of 400 m was applied to prevent samples collected

outside the sampling domain (e.g., samples collected in transit to and from the overnight parking location outside the domain) from being assigned to the nearest magnet in the domain. For comparing measurements made at different sampling intervals (AMS: 20 s; BC: 1 min; CO, particle number: 1 s), we created synthetic 1 Hz AMS and BC datasets such that the measured value at the mid-point of the averaging interval was applied to all 1 second timestamps in that interval. Naturally, for 1 Hz data, the spatial resolution was not limited to 200 m, hence we used magnets spaced every 30 m (same as Apte et al., 2017).

*Unique samples.* The amount of time spent at a 200 m magnet can be longer than 20 s on days when driving was paused at that magnet e.g., for traffic lights, refueling stops, etc. These samples can bias a magnet's representative concentration when averaging is performed across multiple days. Conversely, as shown in the zoom-in (inset D) in Figure 1, a magnet could fall on a route used for transiting from one polygon to another on a particular day. In that case, we treat samples collected at different times of the day as independent, unique samples. This is equivalent to ascribing the same value of information to two data




points collected at different times on a single day as two samples collected on different days. In order to resolve temporally clustered samples from unique samples, we averaged all samples assigned to a magnet within a 60 min window into a single unique sample. For every magnet, the median of all unique samples was chosen as a representative campaign-aggregated measurement.

*Processing AMS data.* We processed AMS data using SQUIRREL 1.57I and PIKA 1.16I routines in Igor Pro 6.37 (Wavemetrics, Lake Oswego, OR; Sueper et al., 2007). We applied three types of corrections to the data: (a) ionization efficiency (IE): two IE calibrations, performed before and after the campaign, provided a two-point estimate of the decay slope of IE over the 20-day period and thus a linearly increasing IE correction factor was applied to the entire AMS dataset; (b) collection efficiency (CE): a composition-dependent CE was calculated using the AMS-measured nitrate fraction in each sample (Mid-

dlebrook et al., 2012); (c) a "zero" offset, obtained daily from concentrations in particle-filtered air while the van was parked outside the sampling domain; signals recorded while sampling particle-free air were also used to resolve the very similar-mass $CHO^+$ ($m/z = 29.002$; particle-phase) and $^{15}NN$ ($m/z = 29.003$; gas-phase isotopic nitrogen) ions. Elemental ratios in this study were calculated using the "Improved-Ambient" method, which uses signal intensities of specific ion fragments to correct for biases in elemental hydrogen-to-carbon (H/C) and oxygen-to-carbon (O/C) ratios (Canagaratna et al., 2015).

*Factorization of OA mass spectra.* To identify sources of OA, we applied positive matrix factorization (PMF) to the two-dimensional OA matrix (time series along rows, concentrations of high-resolution organic ions up to $m/z$ 115 along columns). PMF is essentially a bilinear deconvolution algorithm that explains the OA matrix as a linear combination of variable number of static factors and the time series of their contribution to the total OA. We performed PMF using the PMF2.exe algorithm with the ME-2 multilinear engine (Paatero and Tapper, 1994; Paatero, 2007). We explored different results within the factor-resolved

solution space using the PMF evaluation toolkit (Ulbrich et al., 2009).

    *Accounting for temporal trends.* Over the course of mobile sampling, the urban background air quality can have daily and diurnal variations due to meteorological changes. These variations can be accounted for with the help of concurrent stationary measurements performed at an urban background location. As discussed in Section A1, accounting for temporal trends only had a minor ($\sim 5\%$) effect on the results. We do not include these corrections in the results presented in this manuscript.

*Bootstrap resampling.* In order to compare observations of OA and its factors across areas influenced by different emissions (Port, West Oakland and downtown), we first determined the precision of these measurements by resampling the spatially-aggregated pool of data occurring in these areas. The strength (i.e., number of samples) of a bootstrapped dataset was the same as the strength of the dataset collected in that area. Bootstrapping was performed with replacement and was repeated $10^4$ times. We use the 95% confidence interval of the median as the precision of our measurements. Spatial differences larger than this

precision are then deemed statistically significant. We are not aware of any prior studies that report precision of OA factors.

## 3   Results and discussion

In Figure 2, we show three overall results. First, we show the median aggregated OA concentrations at each magnet. OA is spatially variable, with the highest concentrations typically observed downtown. Second, the spatial coverage of the AMS





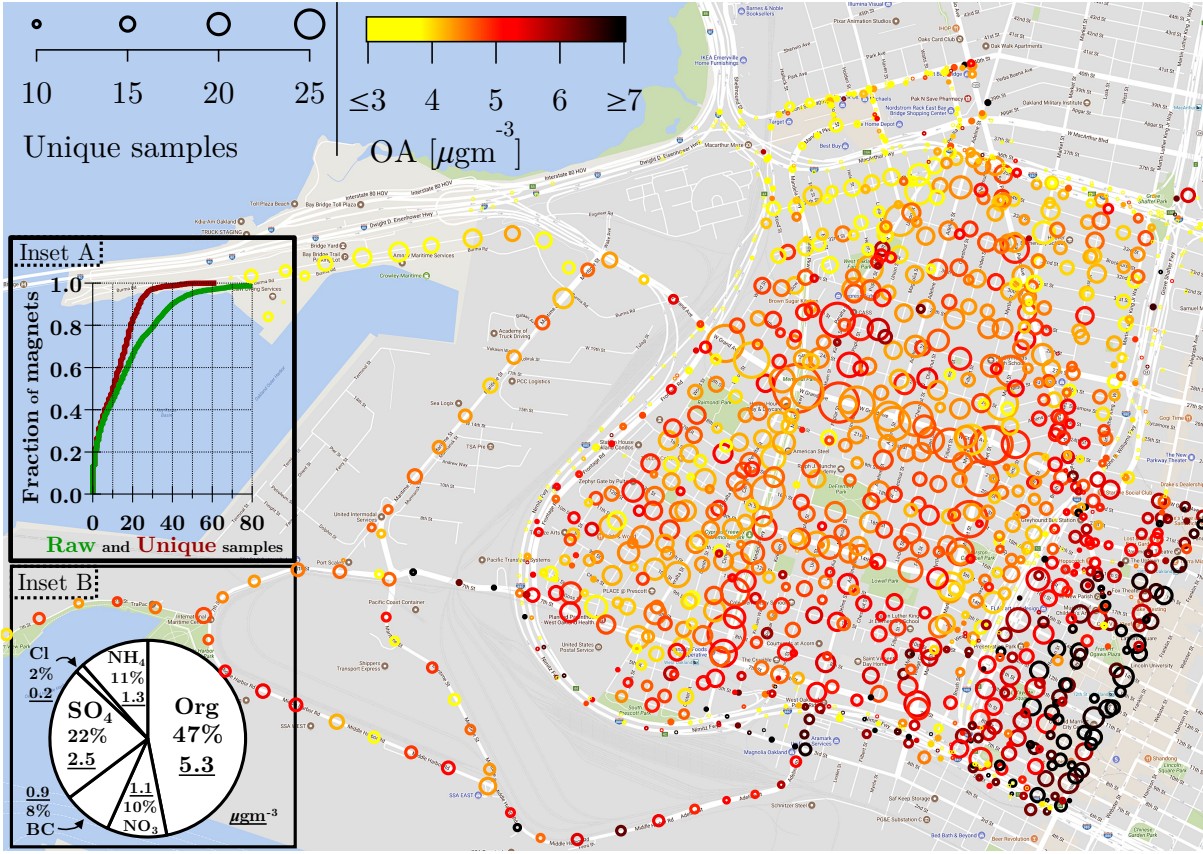

**Figure 2.** Median organic aerosol concentration at each magnet and sampling coverage map of the domain. Inset (a): Cumulative distribution of raw and unique samples in all magnets in domain. Inset (b): median contribution of AMS-measured non-refractory [organics (Org), sulfate ($SO_4^{2-}$), nitrate ($NO_3^-$), ammonium ($NH_4^+$), chloride ($Cl^-$)] and aethalometer-measured black carbon (BC) to the total $PM_1$. Underlined values are absolute mass concentrations in $\mu gm^{-3}$.

dataset is expressed using marker sizes as a function of number of unique samples collected at each magnet. Further, inset A shows cumulative distributions of raw and unique samples across the entire range of magnets. Approximately 60% of the magnets are represented by more than 10 unique samples, suggesting that our data are indicative of long-term spatial patterns (Apte et al., 2017).

5    Lastly, inset B in Figure 2 shows the campaign-median contributions of organics (Org), sulfate ($SO_4^{2-}$), nitrate ($NO_3^-$), ammonium ($NH_4^+$), chloride ($Cl^-$) and black carbon (BC) to the total $PM_1$. The clear dominance of organics along with relative contributions of other species are similar to previously published AMS measurements performed in other urban areas (Hayes et al., 2013; Ortega et al., 2016; Mohr et al., 2015).





## 3.1 Organic aerosol (OA)

We now discuss the spatial patterns of OA in more detail. Figure 3 compares the OA concentrations across the three areas (Port, West Oakland and downtown) using cumulative distribution function (CDF) curves in the upper panel. OA concentrations are spatially variable within each sampled area. A range of $> 2\,\mu\mathrm{gm}^{-3}$ is observed in the median OA concentrations at all magnets of each area.

The lower panel of Figure 3 shows the central tendency statistics (mean, median and standard deviation) of the values assigned to magnets in each area. The data are positively skewed in all polygons i.e., the mean is higher than the median. Ambient measurements typically exhibit a positively-skewed distribution under the influence of local emission events (Apte et al., 2017; Van den Bossche et al., 2015; Brantley et al., 2014). Hence, the median is chosen over the mean as a central tendency statistic to discuss the OA spatial patterns. The results shown in this figure are reinforced with statistical confidence by using bootstrap resampling (Figure S4). We determined a precision of 0.5 $\mu\mathrm{gm}^{-3}$ in the median OA. Spatial differences larger than this precision are considered statistically significant.

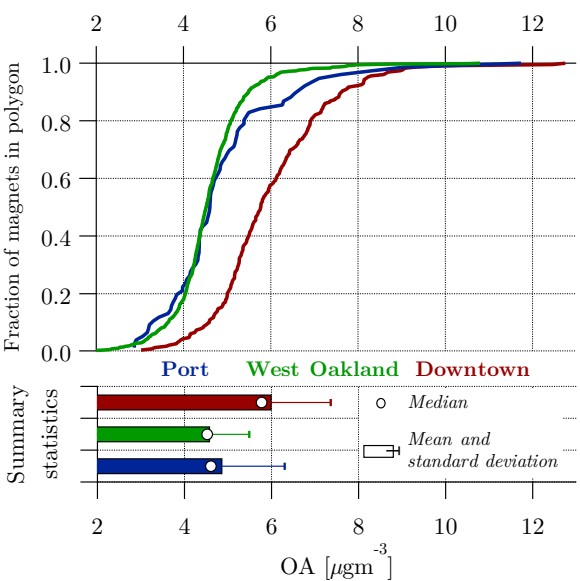

**Figure 3.** Cumulative distributions, mean and median of OA concentrations in Port, West Oakland and downtown.

Downtown has a median OA concentration of 5.7 $\mu\mathrm{gm}^{-3}$, which is 27% higher than West Oakland and Port. Almost the entire downtown CDF curve is $\sim 1.5\ \mu\mathrm{gm}^{-3}$ greater than the West Oakland and Port curves, indicating that all parts of downtown have higher OA concentrations than the rest of Oakland. Port and West Oakland have similar OA concentrations, as evidenced by their nearly-superimposed CDFs and similar medians ($4.6 \pm 0.05\ \mu\mathrm{gm}^{-3}$). However, Port measurements have more positive skewness than West Oakland; Port has a larger fraction of magnets with high OA concentrations than West Oakland. This suggests that the OA concentrations in Port are more influenced by local emission events. As mentioned earlier,




Port has a high drayage truck activity, which likely explains this skewness. Results from bootstrapping support this explanation (Figure S4). Port data have a higher mean (4.9 $\mu$gm$^{-3}$) with a wider 95% confidence interval (0.7 $\mu$gm$^{-3}$) about this mean, compared to West Oakland (mean: 4.6 $\mu$gm$^{-3}$; 95% confidence interval about the mean: 0.2 $\mu$gm$^{-3}$).

## 3.2 Black carbon (BC)

In this subsection, we investigate the influence of higher drayage truck activity on air quality in Port of Oakland. In addition to OA, BC is a prominent component of particulate emissions from heavy-duty diesel trucks (Ban-Weiss et al., 2008; Dallmann et al., 2013).The use of diesel particulate filters and catalytic reducers has substantially reduced BC emissions from drayage trucks in Oakland (Dallmann et al., 2011; Preble et al., 2015). However, Dallmann et al. (2013) reported that diesel trucks contribute 45% of BC concentrations in the Caldecott tunnel in Oakland, despite being a minor ($< 1\%$) fraction of their

sampled vehicle fleet. It is thus reasonable to expect that while the vast majority (99%) of the truck fleet at the Port of Oakland is now equipped with advanced emission control technologies, the overall large volume of trucks arriving in the area results in higher BC emissions than light-duty gasoline vehicles.

We use the OA/BC ratio to distinguish car and truck emissions, with the underlying assumption that these use gasoline and diesel combustion, respectively. By extension, areas with smaller ambient OA/BC would indicate a larger influence from

15 diesel truck emissions. Because OA and BC measurements were made at different sampling frequencies, we compare these measurements by first converting them to a synthetic 1 Hz timeframe as described earlier. A map of median OA/BC values at each 30 m magnet is shown in Figure 4.

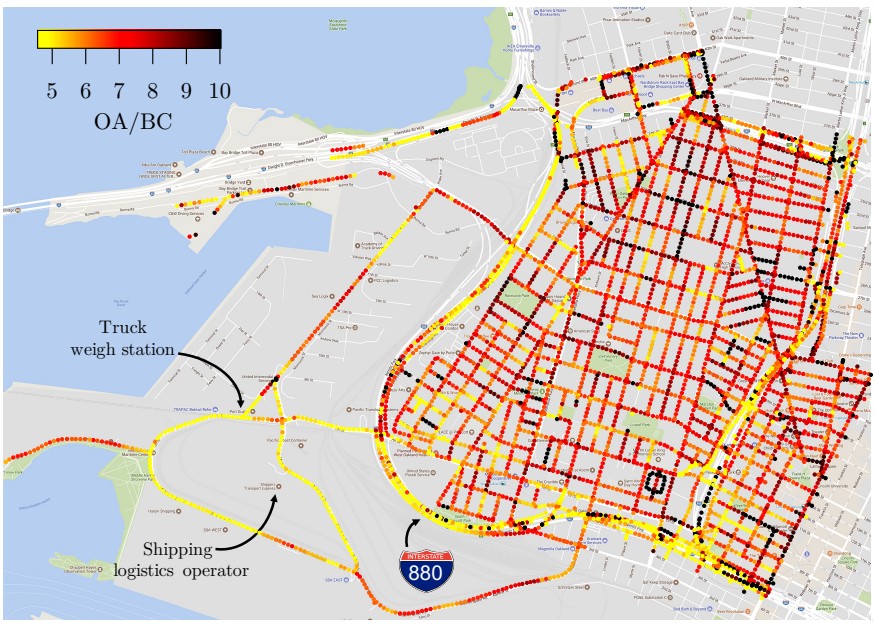

**Figure 4.** Fine-scale map of median OA/BC values (unitless).





In general, areas with lower OA/BC values are streets and highways approaching the Port of Oakland (e.g., Interstate 880 and the surface street connecting it to the port) as well as those around centers of shipping commerce in the Port. This is expected because trucks are found idling while in queue outside these facilities. Bootstrap resampling the BC dataset shows that the measurements (especially in Port) have a considerable positive skew. The ratio of mean to median BC in Port is higher

(1.5) than both West Oakland and downtown (1.2). Because BC concentrations are largely influenced by local sources (diesel trucks), the large amount of drayage trucks in Port likely causes the mean BC to be $\sim 8\%$ higher than downtown, even though median BC in downtown is $\sim 7\%$ higher than Port. This finding suggests that despite substantial reductions in BC emissions from diesel trucks by the use of particulate filters and catalytic reducers, the influence of diesel truck emission plumes on the air quality in Port is higher than that in downtown.

**3.3 OA factors**

We identified three OA factors with distinct mass spectra using positive matrix factorization (PMF) of AMS data: hydrocarbon-like OA (HOA), cooking OA (COA) and semi-volatile oxygenated OA (SV-OOA). These factor profiles are shown in Figure 5. Distinct features of each factor mass spectra (e.g., signals at particular $m/z$'s and elemental O/C ratios) as well as the diurnal patterns in their time series are used to characterize these factors. We use the same nomenclature for these factors as has been

used commonly in literature. For comparison, previously reported mass spectra of these factors are shown in Figure S9.

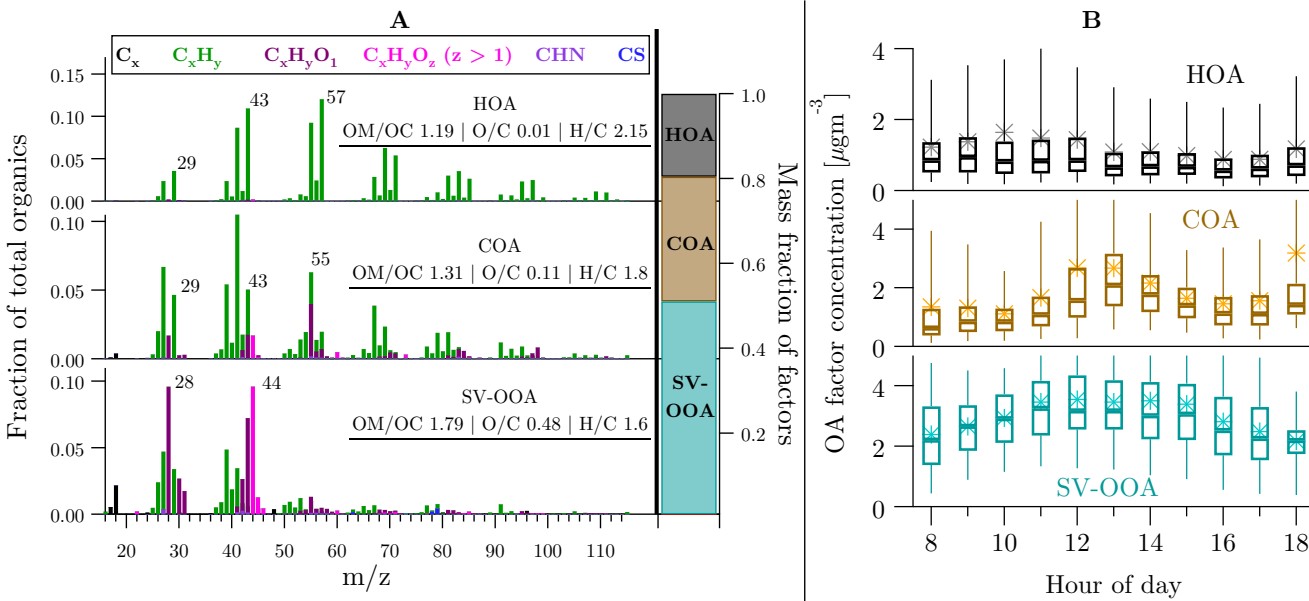

**Figure 5.** (a) Mass spectra, elemental ratios and average mass fraction of factors obtained from PMF analysis of all AMS spectra. (b) Boxplot of diurnal profiles of factors: rectangles enclose 1st through 3rd quartiles of data. Horizontal bars are medians. Asterisk markers are means. Whiskers are 5th and 95th percentiles.





*HOA:* The HOA factor has an elevated signal at the series of $C_nH_{2n+1}$ (e.g., $C_4H_9^+$ at $m/z$ 57) and $C_nH_{2n-1}$ (e.g., $C_3H_5^+$ at $m/z$ 41). Previous studies have identified this factor as a marker of fresh vehicular emissions based on its reduced state (O/C = 0.01) as well as its diurnal pattern (elevated during morning and evening traffic rush periods; Zhang et al., 2011; Mohr et al., 2012). HOA time series are highly positively skewed (average of hourly mean/median = 1.6) even during non-peak periods,
which indicates influence of local HOA plumes.

*COA:* The COA factor has a distinct signal at $m/z$ 55 ($C_4H_7^+$ and $C_3H_3O^+$). Previous studies have identified this factor as a marker of cooking emissions based on its reduced state (O/C = 0.11) as well as its diurnal pattern (elevated concentrations occurring typically during lunch and dinner time; Mohr et al., 2009, 2012; Zhang et al., 2011). Similar to HOA, COA time series are also highly positively skewed (average of hourly mean/median = 1.53), indicating influence of local COA plumes.

*SV-OOA:* Compared to the HOA and COA factors, this factor is relatively more oxygenated with a distinct peak at $m/z$ 44 in its mass spectrum. Secondary OA contains oxygen-containing groups (e.g., carboxylic acids, alcohols and carbonyls). These groups, upon ionization in the AMS, contribute to the $m/z$ 44 ($COO^+$) signal. Generally, based on increasing extent of atmospheric processing (and, by extension, decreasing volatility), two classes of oxygenated OA are identified by signals at $m/z$ 43 and 44: SV-OOA and low-volatility oxygenated OA (LV-OOA) (Donahue et al., 2012; Ng et al., 2010). Of these
two, SV-OOA is relatively less oxygenated and is considered "fresh SOA" formed by gas-phase oxidation of organic precursors emitted nearby (Hayes et al., 2013). SV-OOA is thus found to be strongly correlated with particulate $NO_3^-$.

Having made no thermodenuded measurements of OA volatility, we identify the third, oxygenated factor in our PMF solution as SV-OOA because (a) the mass spectrum and elemental ratios are similar to those reported for SV-OOA elsewhere (Figure S9), and (b) this factor is correlated with AMS-measured $NO_3^-$ signal (Figure S10). The diurnal pattern of the SV-OOA factor
has a more normal distribution about mid-day. Additionally, the SV-OOA time series exhibit less positive skewness (average of hourly mean/median = 1.07) compared to COA and HOA. Compared to the two primary factors, SV-OOA in Oakland has less spatial variability.

*Contribution of OA factors:* Average contributions of COA, HOA and SV-OOA to total OA are shown in Figure 5. COA and HOA collectively contribute ∼ 45% of the total OA mass. This fraction of primary contributions to OA is higher than that
reported for typical urban OA in prior studies (Zhang et al., 2007). Through mobile measurements in Pittsburgh, PA, Gu et al. (2018) find that ∼ 25-30% of the annual average OA mass is primary. The relatively high contributions of primary OA suggest that OA in Oakland is more strongly influenced by local emissions than other locations.

Previous AMS-PMF studies have reported the presence of both SV- and LV-OOA factors in ambient OA. The absence of an LV-OOA factor in Oakland can be explained by the hypothesis that air masses arriving in Oakland are oceanic. These air masses
are expected to contain very low OA concentrations, even though most of this OA is highly oxidized LV-OOA (Hildebrandt et al., 2010). The LV-OOA in these oceanic air parcels would be rapidly overwhelmed by urban emissions as the air parcels are advected over San Francisco and Oakland, resulting in apparent absence of an LV-OOA factor in Oakland. This hypothesis of predominantly oceanic air masses is confirmed by the wind rose diagrams in Figure S16. Predominant winds measured at the Oakland anemometer during periods of mobile sampling are from between the north-west and south-west with typical
wind speeds of ∼ 10 kmh$^{-1}$. This means that Oakland falls roughly 60 min downwind from the Pacific Ocean. Emissions



from the metropolitan SF area are likely advected to Oakland, although the timescale of this advection is <60 min. On this timescale, SV-OOA formation from the SF emissions could be expected, but LV-OOA formation in amounts such that it would be detectable after mixing with the local emissions in Oakland is not expected (Decarlo et al., 2010; Jimenez et al., 2009). The SV-OOA factor profile shown in Figure 5 has a minor contribution from methanesulfonic acid ($m/z$ 79), which was previously

found to be an indicator of marine origin of air parcels (Crippa et al., 2013). This finding confirms that the air masses arriving in Oakland are predominantly oceanic.

*Quality of PMF solution:* PMF decomposes measured OA concentrations using a linear combination of contributions from static factors. The amount of observed mass that cannot be explained by the reconstructed factor contributions is binned into residual mass. Residuals of factorization are shown in Figure S11. The ratio of scaled residuals, $Q$, to the total degrees of

freedom of the fitted data, $Q_{exp}$ would be $\approx 1$ in a perfect factorization (Ulbrich et al., 2009). PMF numerically approaches a convergence using different initial starting points ($f_{peak}$) in the rotational domain about zero (Paatero and Tapper, 1994). Values of $Q/Q_{exp}$ for different values of $f_{peak}$ are shown in Figure S14, along with the factor mass fractions for each solution. Our 3-factor PMF solution is very stable ($1.42 < Q/Q_{exp} < 1.43$) and the factor mass fractions do not change with varying $f_{peak}$.

A 4-factor solution was also examined (Figure S12). While the mass spectra and fractional contributions of both HOA and COA remain unchanged from the 3-factor solution, the SV-OOA factor from the 3-factor solution was further deconvolved into a more oxygenated LV-OOA factor and a fourth less oxygenated factor that bore no similarity to the typical SV-OOA factor spectra reported in the literature. We discarded this 4-factor solution because (a) given that fresh OA factors (HOA and COA) as well as OOA factors form a continuum of atmospheric oxygenation, we do not expect the presence of the fresh OA

and LV-OOA factors while an SV-OOA factor is absent, (b) we did not find a strong PMF-independent tracer correlation (e.g., with AMS-measured particulate $NO_3^-$ or $SO_4^{2-}$) for the LV-OOA and the fourth factor, and (c) going from 3-factor to 4-factor solution, there was only a 5% reduction in $Q/Q_{exp}$, suggesting not only diminishing returns with number of factors $> 3$, but simply an artificial splitting of the optimal solution, which would result in an overinterpretation of the PMF results (Ulbrich et al., 2009). This artificial splitting is also evidenced and discussed later using elemental analysis in Section 3.5.

## 3.4   Spatial and temporal variability of OA and its factors

In this section, we further analyze the spatial and temporal patterns of OA and its factors. We begin with examining the primary-secondary split of OA and how this split varies across space and time. Understanding variability in the primary fraction of OA is important because we know from recent findings that in close proximity to sources such as highways (Saha et al., 2018b), and restaurants (Robinson et al., 2018), there is high amount of primary OA mass in Aitken-mode particles ($< 100$ nm diameter;

Ye et al., 2018) and the particle population is externally-mixed. Atmospheric processing of these emissions (e.g., increasing SOA fraction, coagulation with background particles) makes the particle size distributions more unimodal and internally-mixed in the accumulation mode (200-1000 nm diameter range; Ye et al., 2018). This primary-secondary split may have important health exposure implications because Aitken-mode particles have longer retention times once they penetrate lung tissue (Ferin et al., 1992; Oberdürster, 2000; Stölzel et al., 2007).





Overall, the OA mass in Oakland is split into primary and secondary factors roughly evenly (Figure 5): two primary factors (COA + HOA) collectively contribute $\sim 45\%$, while the rest is secondary (SV-OOA). However, the primary/secondary split at each magnet is spatially and temporally variable. To identify areas with elevated levels of primary emissions, we normalized the sum of COA and HOA to the total OA concentration at each magnet, which results in a primary fraction of the OA at the magnet. In Figure 6A, we show a map of primary fraction of OA at each magnet in the domain. It is evident that the OA in parts of downtown has a higher than 50% contribution from primary sources. Other areas exhibiting higher fractions of primary sources are the highways. In the more residential areas (West Oakland), SV-OOA contributes $\sim 55\%$ of the OA (Figure 6D). Through bootstrap resampling, a precision of $\sim 1.5\%$ was determined for the primary fraction of OA (Figure S6).

Figure 6B shows a map of the SV-OOA mass concentrations. While SV-OOA has a smoother spatial pattern than primary OA, downtown exhibits higher SV-OOA concentrations. Individual maps of COA and HOA are shown in Figure S17. Figure 6 also shows the diurnal profiles of the primary contributions to OA in Port, West Oakland and downtown. The primary OA contributions in Port and downtown are more temporally variable than in West Oakland. Generally, primary OA contributions in Port have a positive trend (from $\sim 25\%$ to 60%) with increasing hour of day, with peaks occurring at $\sim 11$ AM and 6 PM. This is likely due to increasing volume of trucks as the day progresses. By contrast, primary contributions in downtown are consistently around 50% of the total OA, with peaks during high traffic periods and lunch time. Primary contributions in West Oakland are consistently around 45%, with no distinct peaks.





**Figure 6.** (a) Median primary fraction of OA at each magnet. Primary fraction is defined as the ratio of COA + HOA to the total OA at each magnet. (b) Median SV-OOA concentration at each magnet. Also shown are diurnal profiles of the primary OA fraction in (c) Port, (d) West Oakland, and (e) downtown. Solid and dashed lines are hourly medians and means, respectively. Darkly-shaded areas enclose 1[st] and 3[rd] quartiles of data. Lightly-shaded areas enclose 5[th] through 95[th] percentiles of data.

Figure 7 shows a scatter plot of COA versus HOA contributions to OA from the entire campaign, which is meant to show the diurnal variation of primary OA sources. During the morning rush hour, concentrations of COA and HOA are roughly equal.




Primary OA concentrations start transitioning towards COA dominance at $\sim$ 11 AM, which co-incides with the typical time restaurant kitchens would be expected to begin activity. By $\sim$ 2 PM, COA dominates the primary OA mass.

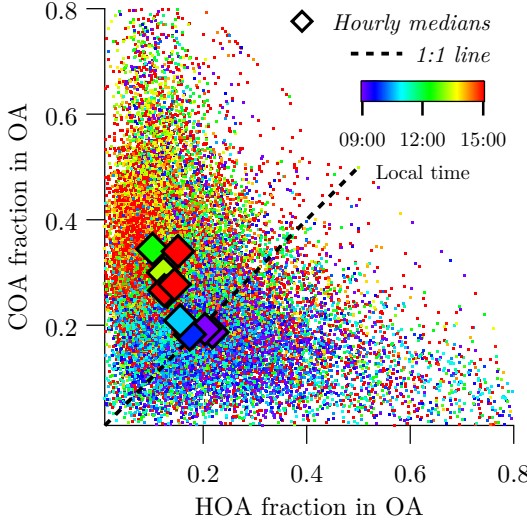

**Figure 7.** Diurnal shift of source of primary OA from morning rush hour traffic to mid-day cooking activities. Dots show individual measurements and diamonds show hourly medians over the entire sampling campaign.

In Figure 8, we show the spatial and temporal patterns of total OA as well as the three OA factors by resolving them by area (Port, West Oakland and downtown) and time of day: morning (8 to 11 AM), mid-day (11 AM to 2 PM) and afternoon
5  (2 to 6 PM). To aid discussion with statistical confidence, we refer the reader to Figure S7, which shows 95% confidence intervals of these results obtained from bootstrap resampling. As described earlier, we consider the 95% confidence interval of the bootstrapped median as an indicator of precision i.e., spatial differences larger than this precision are deemed statistically significant. Overall, OA factors have a $\sim$ 0.25 $\mu$gm$^{-3}$ precision. This precision agrees with that determined in an independent dataset acquired in Pittsburgh, PA (Gu et al. *in prep.*).





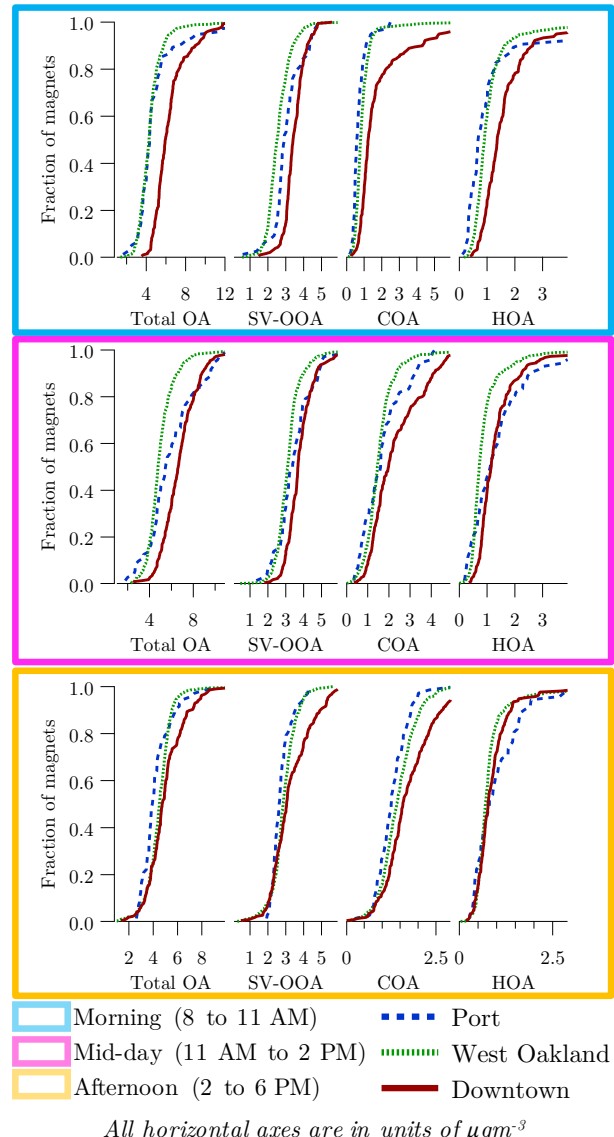

**Figure 8.** Cumulative distribution functions of total OA as well as the three factors identified in this study, resolved by area (Port, West Oakland and Downtown) and time of day: morning (8 to 11 AM), mid-day (11 AM to 2 PM) and afternoon (2 to 6 PM). The colored box around each set of CDFs indicates the period of the day those data represent.

*OA.* During the morning period (8 to 11 AM), the median OA concentration in downtown is 40% ($\sim 1.7 \ \mu \mathrm{gm}^{-3}$) higher than in West Oakland and Port. Median OA in Port and West Oakland are similar (within precision). On average, the median OA concentration in the entire domain is 16% ($\sim 0.8 \ \mu \mathrm{gm}^{-3}$) higher during mid-day (11 AM to 2 PM), compared to the morning period. Median OA concentration in downtown is 40% ($\sim 1.9 \ \mu \mathrm{gm}^{-3}$) higher than in West Oakland and 21% higher than in Port. During the afternoon (2 to 6 PM), the median OA concentration over the entire domain is 20% ($\sim 1.2 \ \mu \mathrm{gm}^{-3}$) lower





than the mid-day period. Median OA concentration in downtown is similar to West Oakland ($4.6 \pm 0.14$ $\mu$gm$^{-3}$) and 17% higher than in Port. Overall, total OA concentrations in downtown are consistently higher than Port and West Oakland during morning, mid-day and afternoon periods.

*HOA.* During the morning period, median HOA concentration in downtown is 52% ($\sim 0.5$ $\mu$gm$^{-3}$) higher than in West

Oakland. Median HOA concentration in West Oakland, in turn, is 32% higher than Port. There is no net increase in median HOA concentration in the entire sampling domain from morning to mid-day. However, from morning to mid-day, the median HOA reduces by 16% and 20% in downtown and West Oakland respectively, while that in Port increases by 64% ($\sim 0.4$ $\mu$gm$^{-3}$). As a result, during mid-day, the median HOA in downtown and in Port are similar ($1.1 \pm 0.01$ $\mu$gm$^{-3}$), and higher than that of West Oakland by 58% ($\sim 0.4$ $\mu$gm$^{-3}$). On average, the median HOA concentration in the entire domain is $\sim 0.2$

$\mu$gm$^{-3}$ lower during the afternoon compared to the mid-day and morning periods. Median HOA concentrations in the three polygons are all similar $0.8 \pm 0.06$ $\mu$gm$^{-3}$, with Port being highest.

  Overall, HOA is highest in downtown in the morning, despite the fact that all of Oakland has roughly equal proximity to highways. While we do not have detailed traffic data for Oakland, it is reasonable to assume that downtown receives a large influx of commuters during the morning rush hour and thus would be expected to have the highest HOA concentrations.

Downtown also has a higher road length density compared to West Oakland and Port, and as a result, can accommodate a larger traffic volume per km$^2$. Further, poor ventilation of vehicle emissions in downtown can also contribute to higher HOA levels (Yuan et al., 2014). During mid-day and afternoon, however, Port has similar or higher HOA than downtown. This is expected for two reasons: (a) the commuters contributing to high HOA in downtown during the morning hours are working (and their cars parked) during midday, and (b) the high amount of drayage truck activity in Port. There is some evidence from fuel sales

data that truck activity at the Port is higher in the afternoon than the morning. One of the measures implemented to reduce Port emissions was to keep shipping logistics gates open in the evening to dilute daytime congestion of drayage trucks (Port of Oakland, 2016), which may contribute to trucks at the Port not following the typical rush hour traffic patterns of commuters. West Oakland has the lowest HOA concentrations at all times of day, as would be expected for a largely residential area.

  *COA.* Median COA concentration in downtown is 55% ($\sim 0.5$ $\mu$gm$^{-3}$) higher than in West Oakland and 109% higher than

in Port during the morning period. Further, COA concentrations in downtown exhibit a larger positive skew (mean/median = 1.5) relative to both West Oakland and Port (mean/median = 1.15). On average, the median COA concentration in the entire domain is $\sim 0.8$ $\mu$gm$^{-3}$ higher during mid-day, compared to the morning period. Median COA concentration in downtown is 27% higher than in West Oakland and Port during mid-day. On average, the median COA concentration in the entire domain is $\sim 0.3$ $\mu$gm$^{-3}$ lower during the afternoon compared to the mid-day period. Median COA concentration in downtown is 11%

and 26% higher than in West Oakland and Port, respectively.

  Overall, COA is consistently highest in downtown, which is not surprising given the large number of restaurants in downtown. The spatial distribution of COA in West Oakland and Port is uniformly low in the morning, as demonstrated by the steepness of the distribution functions in Figure 8. During mid-day and late afternoon, however, COA levels in both these areas are higher than in the morning. It is not clear why COA in Port is higher than WO during mid-day. It should be noted, however,

that due to the considerably low road length density in Port, the number of magnets in Port is only $\sim 10\%$ of those in West



Oakland. As a result, even a few cooking sources in Port (most likely food trucks) could cause the spatially aggregated values for Port appearing to be higher than West Oakland. Overall, Port has the lowest COA concentrations, which is expected given the land use in that area.

*SV-OOA.* Median SV-OOA concentration in downtown is 36% ($\sim 0.9$ $\mu$gm$^{-3}$) and 17% higher than in West Oakland and Port, respectively, during the morning period. On average, the median SV-OOA concentration in the entire domain is 15% ($\sim 0.5$ $\mu$gm$^{-3}$) higher during mid-day than the morning period. Median SV-OOA concentration in downtown is 18% ($\sim 0.6$ $\mu$gm$^{-3}$) higher than in West Oakland and 8% higher than in Port. On average, the median SV-OOA concentration in the entire domain is 20% ($\sim 0.5$ $\mu$gm$^{-3}$) lower in the afternoon, compared to the mid-day period. Median SV-OOA concentrations in all three areas are similar ($2.9 \pm 0.2$ $\mu$gm$^{-3}$), with downtown being the highest and Port being lowest.

SV-OOA concentrations are consistently higher in downtown compared to Port and West Oakland. This finding is unexpected because SV-OOA is secondary; the null hypothesis for secondary species is that concentrations would be spatially uniform. Multiple lines of evidence contribute to the conclusion that SV-OOA is indeed higher in downtown than the Port and West Oakland. First, as shown in Figures 8 and S7, morning-time SV-OOA concentrations are 0.9 and 0.5 $\mu$gm$^{-3}$ higher in downtown than in Port and West Oakland. Given our determined precision of 0.25 $\mu$gm$^{-3}$ in OA factors, these spatial differences are significant. As discussed above and shown in Figure S1, our sampling times are not temporally biased; therefore, higher SV-OOA downtown does not appear to be an artifact of our sampling strategy (e.g., downtown is not oversampled at midday relative to West Oakland and Port). Second, the SV-OOA CDFs in Figure 8 are further substantiated by East-West transect drives performed during inter-polygon transits on different days of the campaign and shown in Figure 9. These transects show that there is a general trend of increasing SV-OOA concentrations between the Port and downtown. Third, the enhanced SV-OOA downtown echoes similar measurements made independently in Pittsburgh, where fresh SOA is also enhanced in the downtown area (Gu et al. *in prep.*).

Higher concentrations of SV-OOA in downtown suggest that there is enhanced photochemical activity in that area. Enhanced photochemical production of SOA could be due to several reasons. Firstly, the pool of reactive SOA precursor vapors is likely enhanced in downtown relative to other areas. We show above in Figure 8 that HOA is highest in downtown. HOA is co-emitted with volatile and intermediate-volatility organic compounds that efficiently generate SOA upon photochemical oxidation (Robinson et al., 2007; Gordon et al., 2013; Zhao et al., 2015).

Secondly, concentrations of the hydroxyl (OH) radical may also be higher downtown. The OH radical is the dominant daytime oxidizing agent, especially for reduced compounds emitted from motor vehicles. It has been shown that in urban, polluted environments (street canyons), high-NO$_x$ emissions from vehicles render increased levels of nitrous acid (HONO), which in turn results in increased local oxidation capacity (Yun et al., 2017; Villena et al., 2011) due to rapid photolysis of HONO to OH (Stutz et al., 2000; Finlayson-Pitts and Pitts, 2000; Zhong et al., 2017).

A third factor may also contribute to higher precursor concentrations, and therefore additional SV-OOA, in downtown. The presence of tall buildings in downtown can create higher surface roughness, which in turn can reduce pollutant dispersion and promote internal recirculation (Zhong et al., 2017). This possibility is further examined in Figure S18, which shows building heights in downtown. Downtown has relatively taller buildings that collectively act as a wall parallel to the Interstate 980





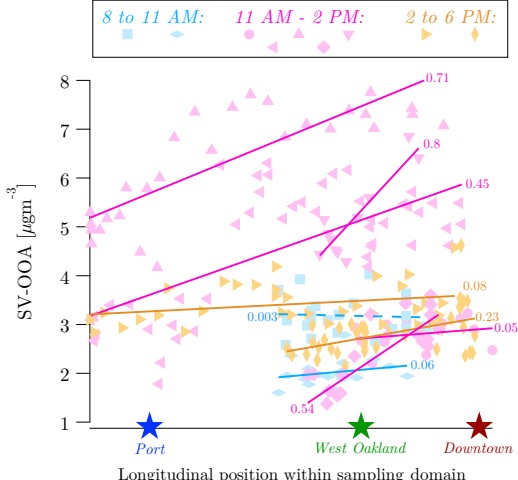

**Figure 9.** Spatial variations in measured SV-OOA on select intra-polygon transit drives on different days, colored by three different diurnal time periods. Drives that reasonably spanned the longitudinal extent of the sampling domain with minimal latitudinal displacement were picked for this plot. Markers are individual SV-OOA samples and lines are fits for each transect drive. Fits that have a positive slope with proximity to downtown are shown as solid lines. The only fit that has a negative slope is shown as a dashed line. $R^2$ of fits are shown next to fits. Approximate locations of Port, West Oakland and downtown are shown as references to better visualize longitudinal positions of data.

highway upwind. The air masses in downtown may experience stagnation and poor ventilation relative to West Oakland and Port. This, in turn, can increase the reaction time, thereby allowing more local SV-OOA formation.

The combined impact of vehicle emissions on OH (via HONO) and gas-phase precursor concentrations in downtown would be expected to be largest in the morning, since these are co-emitted with HOA. Indeed, the largest enhancement of SV-OOA

in downtown occurs in the morning hours, at the same time as the largest enhancement of HOA. From morning to mid-day, SV-OOA concentrations become more spatially uniform; median SV-OOA concentrations in West Oakland and Port increase by 23% (0.6 $\mu gm^{-3}$) and 16% (0.5 $\mu gm^{-3}$) respectively, while downtown only increases by 7% (0.25 $\mu gm^{-3}$). This suggests that while downtown has high photochemical activity in the morning, the entire sampling domain transitions towards a more uniform photochemical state by mid-day.

We further investigated the enhanced photochemical activity in downtown by analyzing mobile measurements of particulate sulfate ($SO_4^{2-}$). $SO_4^{2-}$ is formed upon reaction of gas-phase $SO_2$ with OH (Miyakawa et al., 2007), which can be enhanced in polluted environments due to catalytic involvement of black carbon particles (Novakov et al., 1974). Figure S8 shows that $SO_4^{2-}$ concentrations in downtown are higher ($\sim 8\%$) than Port and West Oakland, a trend similar to that observed in SV-OOA concentrations.

Ships associated with the Port are the major source of $SO_2$ in Oakland (Tao et al., 2013). While ship $SO_2$ emissions have significantly decreased in recent years, ships remain the dominant $SO_2$ source in our sampling domain, and there seem to be no local sources of $SO_2$ or particulate $SO_4^{2-}$ in downtown. Thus, elevated concentrations of secondary sulfate in downtown





would therefore support the hypothesis that OH concentrations are also higher in downtown, giving rise to local variations in $SO_4^{2-}$ formation rate.

## 3.5 Elemental analysis of bulk and factor OA

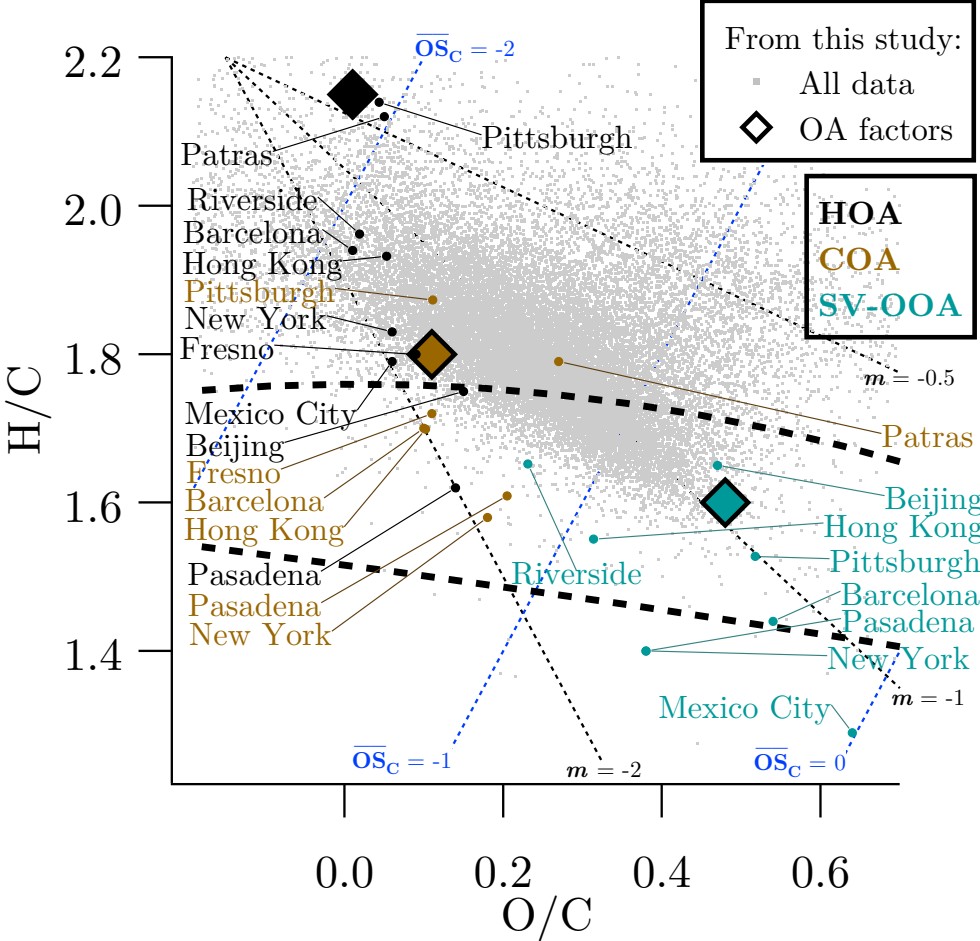

**Figure 10.** The Van Krevelen plane. Gray points represent all OA measurements in this study. Diamonds represent OA factors identified in this study. For reference, placement of OA factors from other ambient measurements are shown: Pittsburgh (Gu et al. *in prep.*), Patras, Greece (Kaltsonoudis et al., 2017), Riverside, CA (Docherty et al., 2011), Barcelona (Mohr et al., 2012), Hong Kong (Lee et al., 2015), New York, NY (Sun et al., 2011), Fresno, CA (Ge et al., 2012), Mexico City (Decarlo et al., 2010), Beijing (Hu et al., 2016) and Pasadena, CA (Hayes et al., 2013). Different oxygenation pathways are shown by the black dotted lines. Dotted blue lines are isopleths for average carbon oxidation states ($\overline{OS}_C = 2 \times O/C - H/C$). The region of ambient oxygenated OA measurements, as reported by Ng et al. (2011), is shown between the dashed curves.





In this section, we investigate the measured elemental ratios (H/C and O/C) of OA using a Van Krevelen (VK) diagram. Figure 10 shows all bulk OA data from this study (gray dots). Only 21% of the bulk data fall inside the typical ambient range of OOA measurements (Ng et al., 2011), while the remaining data are on the less oxygenated side of this range. This indicates a dominant presence of highly reduced, primary OA in Oakland. This finding contrasts with the measurements made in other

urban areas. For instance, almost all of the bulk OA measurements made in Pasadena, CA were either inside or below (i.e., more oxygenated than) this ambient OOA region (Ortega et al., 2016). This finding is consistent with the large contribution of primary OA in Oakland (Figures 5 and 6) and can be partially explained by the overall spatial proximity of ≤ 1 km of all points inside the domain to the nearest highway (Apte et al., 2017). Primary emissions are thus of more importance in Oakland than other urban areas.

Figure 10 also shows that the cluster of bulk OA data from this study aligns with the -1 slope line (slope of linear fit = -1.003, not shown; Pearson's R = 0.62). Previous studies have used VK slopes to determine if the processing of ambient OA is dominated by chemical (e.g., oxygenation) or physical (e.g., external mixing of air masses) mechanisms. By isolating chemical processing of Pasadena OA in an oxidation flow reactor, Ortega et al. (2016) showed that chemical processing shifts OA on the VK plane along a shallow slope of $\sim -0.7$. A similar slope was observed by Liu et al. (2018) by chemical processing

of Beijing OA in an oxidation flow reactor. Similarly, Presto et al. (2014) measured a slope of $\sim -0.5$ for oxidation of fresh gasoline and diesel exhaust in a smog chamber. By comparing OA measurements in Riverside, CA with those in Pasadena, Hayes et al. (2013) hypothesized that the HOA-to-SV-OOA transformation in Riverside occurred with a steeper VK slope ($m = -1.1$) due to the physical mixing of highly reduced ($\overline{OS}_C = 2 \times O/C - H/C = -1.92$; Kroll et al., 2011; Canagaratna et al., 2015), HOA-rich air masses with OOA-rich air masses in Riverside.

Figure 10 also plots the HOA, COA and SV-OOA factors identified in this study on the VK plane. For reference, similar factors are shown from ambient measurements reported in prior publications. As expected, reduced factors corresponding to primary emissions (HOA and COA) occur closer towards the top-left corner of the VK plane, while the SV-OOA factors occur closer towards the bottom-right corner. With the exception of Pittsburgh and Patras, HOA in Oakland is more chemically reduced ($\overline{OS}_C$ = -2.13) compared to HOA in other locations (average $\overline{OS}_C \sim$ -1.7). That the composition of HOA in Oakland

is more reduced, as well as the overall slope for all OA measurements in Oakland is $\sim$ -1, suggests that external mixing of highly reduced, HOA-rich air masses with OOA-rich air masses is an important processing mechanism in Oakland. This is also consistent with the recent findings of Ye et al. (2018) via single-particle mass spectrometry in Pittsburgh, PA; particle mixing state shifts from internal to external with $\sim$ km-scale proximity to urban sources. It is also worth noting in Figure 10 that the HOA measurements that are more chemically-reduced (Oakland, Pittsburgh and Patras) were performed in the past 5

years, compared to the other measurements that were made earlier ($\sim$ 10 years ago). This suggests a change in the chemical composition of vehicular emissions over the past decade.

Finally, this VK analysis also reinforces the choice of a 3-factor PMF solution in this study. In Figure S15, the 3- and 4-factor PMF solutions are compared on the VK plane, along with other reference data as already described in Figure 10. The HOA and COA markers do not change their position on the VK plane between the 3- and 4-factor solutions. This is consistent with

the robustness of HOA and COA mass fractions to the choice of PMF solutions, as described earlier and shown in Figure S14.





However, the SV-OOA factor in the 3-factor solution is split into a highly oxygenated LV-OOA factor and a fourth, unknown factor. We explained earlier that the 4-factor solution is an artificial splitting of the 3-factor solution. The placement of these factors on the VK planes is consistent with this explanation. The LV-OOA factor occurs in the far bottom-right corner, where there are no bulk OA data. The fourth OA factor coincides with the center of mass of the bulk OA data cluster, likely because the artificial splitting of the SV-OOA factor is done with the constraint of minimizing residuals ($Q/Q_{exp}$). Further, the SV-OOA, LV-OOA and the unknown factor all fall on the same $m = 1$ dotted line. Thus, from a strictly mathematical standpoint, this solution offers no new information, given that the LV-OOA and the unknown factor average along the $m = 1$ line to form the SV-OOA factor.

## 4 Summary and conclusions

Having one of U.S.'s largest shipping ports, the air quality in Oakland has been historically impacted by shipping-related activities such as presence of ships burning high-sulfur fuel and drayage trucks driving through the directly adjacent residential neighborhood (Fisher et al., 2006; Fujita et al., 2013). In the past decade, regulations on drayage trucks and ship fuel usage have dramatically reduced emissions (Tao et al., 2013; Dallmann et al., 2011; Preble et al., 2015). Findings of this study are consistent with these previous studies regarding the substantially improved air quality in Oakland. However, the port is not the only source of emissions that could impact the air quality in Oakland. The city also has a central business district ("downtown") that has activities such as domestic vehicular traffic and cooking, similar to other urban areas. Therefore, urban downtowns are also a prominent source of organic aerosols (OA; Mohr et al., 2015), which make up a dominant part of particulate matter (PM). Since the residential West Oakland neighborhood falls in the middle of the port and downtown, it is important from a health exposure perspective to determine the spatial variability of pollutants within the city and to determine which of these two area sources drive spatial variability in PM.

The objective of this study was to examine the spatial and temporal patterns in pollutants impacting the air quality in Oakland through mobile sampling in an instrumented van. Organic aerosol (OA) contributes the largest fraction ($\sim 50\%$) of PM$_1$ mass. We find that primary emissions from cooking (COA) and vehicles (HOA) as well as secondary OA (SV-OOA) contribute to the spatial variation within Oakland. Key findings are:

1. Organic aerosol is the dominant component of PM$_1$ (OA; $\sim 50\%$), and its contribution is roughly twice that of sulfate (23%). This finding is consistent with the that of Tao et al. (2013), who also showed that the enforcement of low-sulfur fuel usage in ships has successfully reduced the amounts of particulate-phase $SO_4^{2-}$ in the local PM in Oakland.

2. In downtown, concentrations of primary OA are higher than secondary OA. The dominant source of these primary OA emissions shifts diurnally between cooking and vehicles: pre-10 AM fresh emissions are from vehicles, but cooking emissions contribute dominantly to OA after lunchtime. Further, by investigating the ratio of OA emitted from vehicles to black carbon (BC; particulate matter emitted typically from diesel combustion in trucks), we show that drayage truck emissions do affect the spatial variability of OA in space (Port) and time (afternoon, when truck traffic typically peaks).



3. Secondary OA (SOA) also exhibits spatial variability similar to primary OA. SV-OOA concentrations are higher in downtown, likely because a combination of various factors (poor ventilation of air masses in street canyon, higher emissions of SOA precursors, higher OH concentrations) result in downtown being a microenvironment with high photochemical activity.

4. Overall chemical composition of OA in Oakland is more chemically-reduced relative to that reported in studies in several other locations. Especially, HOA in Oakland is more reduced than other locations. This reflects the importance of primary emissions in Oakland. Further, by comparing measurements from other studies, we show that the chemical composition of HOA has likely become more reduced over the last decade. Lastly, external mixing of air masses with contrasting pollutant concentrations plays an important role in the processing of these chemically-reduced emissions.

These findings have important implications for population exposure studies because urban downtowns tend to have a concentrated presence of workplaces, resulting in people being exposed to these elevated pollutant concentrations for ∼ 8 h (typical work day duration) everyday. These findings are experimentally shown for Oakland as a case city, which has unique elements (oceanic winds mixing with urban emissions, presence of a major port) that likely have a unique influence on the intra-city patterns in its air quality. However, Oakland also has elements that are fairly typical of urban areas (e.g., downtown with high cooking and vehicular emissions). The findings of this study are thus likely applicable to other urban areas as well, because the pollutants we find contributing the most to OA variability, both of primary and secondary origin, are ubiquitous in other urban locations.

*Author contributions.* RUS, ESR and PG collected data. RUS performed data analysis with input from all co-authors on the interpretation of results. RUS and AAP wrote the manuscript with significant input from ALR. ALR, JSA and AAP designed the research.

*Competing interests.* The authors declare no competing financial interest.

*Acknowledgements.* This research was funded by Environmental Defense Fund (EDF) and NSF grant number AGS1543786.
This publication was developed under Assistance Agreement No. RD83587301 awarded by the U.S. Environmental Protection Agency. This work has not been formally reviewed by the funding agencies. The views expressed in this document are solely those of the authors and do not necessarily reflect those of the funding agencies. EPA does not endorse any products or commercial services mentioned in this publication.
We thank Sarah Seraj (UT Austin) for her assistance with data collection; Thomas Kirchstetter, Chelsea Preble and Julien Caubel (Lawrence Berkeley National Lab) for assistance with calibration instrumentation.



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
