# Peer review of "High spatial resolution mapping of aerosol composition and sources in Oakland, California using mobile aerosol mass spectrometry"

_Atmospheric Chemistry and Physics, 2018_

## Referee Comment (RC1) · Anonymous Referee #1 · 24 Jul 2018

This manuscript shows high-resolution spatial patterns of PM composition and amount measured by an AMS on a truck in Oakland. Overall, I found the paper to be well within the scope of ACP, the analysis was well done, and the paper was well written.

I have several comments that I'd like addressed before publication in ACP.

General comment (because it shows up in a few places):

Bottom $\frac{1}{3}$ of P18, but also the abstract and point #3 on P23: To me, the simplest explanation as to why SV-OOA is higher in downtown is that it's the location that's farthest downwind (wind typically heading from west to east in Figure S16). The mid-day gradient in SV-OOA appears to be steadily increasing from west to east in Figure

9, rather than a step change to higher values when entering downtown. The air starts picking up SOA precursors when it first hits land (either on the west or east sides of the bay) and SOA forms as the air moves across Oakland. Certainly more precursors are being added in Downtown, which helps, but the air as simply had more time to make SOA from the precursors (or oxidize the HOA and COA) by the time the air reaches downtown.

The proposed hypotheses in the manuscript that deal with enhanced photochemical activity seem like less straightforward explanations. Downtown might be in a high-NOx regime, which would lower OH (acting against the proposed HONO source).

Unless I'm missing something, I don't know why the simple "amount of time spent over land" hypothesis isn't prominent in the paper.

Specific comments:

Abstract: The final sentence is subjective and unnecessary, in my opinion. The paper shows many things that would seem to be unique to Oakland (in addition to things that are likely common with other urban locations).

P2 L14-15: The length and time scales don't seem to match. Winds would need to be very stagnant («1 km/hr) in order for "hundreds of meters" to correspond to "hours".

P3 L7: need a comma after "legislations"

P3 L10: I had to look up what "drayage" meant, and I suspect others might not know as well. Since the word is used quite a bit in the paper, it may be worth giving a very brief explanation here.

P6 L25-30: I know, generally, what bootstrapping statistical methods are, and I've used some. However, there was not enough info in this paragraph to really understand what was done (I could not repeat the analysis based on this description).

---

## Referee Comment (RC2) · Anonymous Referee #2 · 30 Aug 2018

General comments

This paper presents a detailed study on aerosol composition, focusing on the organic components, in Oakland, California. A mobile laboratory carrying an aerosol mass spectrometer (HR-ToF-AMS) was used to map the city roads in about 160 hours of measurements, resulting in a highly spatial resolved data set. The organic aerosol was separated using PMF into HOA, COA, and SV-OOA. While the manuscript is technically very sound and presents the AMS data analysis in great detail, it falls short in the interpretation of the results. To my opinion, more could have been done (or has to be done) in this direction, before publication in ACP can be recommended. My detailed

comments are listed in the following.

Major comments

1) As written above, the interpretation of the results is not satisfying. That a large city like Oakland has a strong HOA influence would have been expected. The authors emphasize that Oakland has one of the largest shipping ports in the US. Thus, I would have expected to see as a result a separation of the HOA influence between ship exhausts, port-related traffic (trucks) and individual car traffic. I do not find any of this in the results or conclusions sections, only a little of such an analysis in section 3.2. The title suggests that not only the spatial distribution but also the sources of aerosol particles are investigated. It would have been interesting to learn about the reduction potential for the PM burden in a city like Oakland, e.g. whether reduction of ship emissions, truck or car traffic have a higher impact on PM loads.

2) The data analysis is restricted to AMS and black carbon data. But, as was written in section 2.2, also CO, CO2 and particle number concentration was measured. Apparently neither CO2 nor particle number were used here in the analysis. No correlation between CO and HOA is presented. I think that the addition of these parameters (e.g. ratios OA/CO, correlations between BC and CO, HOA and CO, HOA and particle number. . .) would be a benefit for the analysis.

Minor comments

Page 2 line 24-25: Also in Paris (MEGAPOLI) COA was identified to be of high importance (Crippa et al., 2013; Freutel et al., 2013)

Page 7, Figure 2: The figure suffers from too much information. I suggest adding an extra figure in the Supplement with the number of unique samples and have in this figure only filled symbols color coded for OA.

Page 8, lines 8.9: "Ambient measurements typically exhibit a positively-skewed distribution under the influence of local emission events." Really? Are three papers enough

to say "typically"? And why is that so?

Page 10, lines 11-15: It was suggested by Sally Ng's group (e.g., Xu et al., 2015) to replace "LV-OOA" and "SV-OOA" by "More-oxidized and less-oxidized oxygenated organic aerosol (MO-OOA and LO-OOA)". You may consider using LO-OOA instead of SV-OOA.

Page 12: lines 0-5: Oceanic air masses: There is literature on marine OOA factors (e.g. Ovadnevaite et al., 2011; Schmale et al., 2013). There is some potential in this marine influence on aerosol properties. You could make more of it.

Summary and conclusions: As already mentioned above, this section is very short. I think a statement on the relative influence of ships at the port, trucks at the port, and individual car traffic on the aerosol burden in the city would have been the desired output of this study.

Technical

Page 6, line 22: Link to appendix is missing

Page 12, line 34: Oberdörster

References:

Crippa, M., DeCarlo, P. F., Slowik, J. G., Mohr, C., Heringa, M. F., Chirico, R., Poulain, L., Freutel, F., Sciare, J., Cozic, J., Di Marco, C. F., Elsasser, M., Nicolas, J. B., Marchand, N., Abidi, E., Wiedensohler, A., Drewnick, F., Schneider, J., Borrmann, S., Nemitz, E., Zimmermann, R., Jaffrezo, J.-L., Prévôt, A. S. H., and Baltensperger, U.: Wintertime aerosol chemical composition and source apportionment of the organic fraction in the metropolitan area of Paris, Atmos. Chem. Phys., 13, 961-981, https://doi.org/10.5194/acp-13-961-2013, 2013.

Freutel, F., Schneider, J., Drewnick, F., von der Weiden-Reinmüller, S.-L., Crippa, M., Prévôt, A. S. H., Baltensperger, U., Poulain, L., Wiedensohler, A., Sciare, J., Sarda-

Estève, R., Burkhart, J. F., Eckhardt, S., Stohl, A., Gros, V., Colomb, A., Michoud, V., Doussin, J. F., Borbon, A., Haeffelin, M., Morille, Y., Beekmann, M., and Borrmann, S.: Aerosol particle measurements at three stationary sites in the megacity of Paris during summer 2009: meteorology and air mass origin dominate aerosol particle composition and size distribution, Atmos. Chem. Phys., 13, 933-959, https://doi.org/10.5194/acp-13-933-2013, 2013.

Ovadnevaite, J., C. O'Dowd, M. Dall'Osto, D. Ceburnis, D. R. Worsnop, and H. Berresheim (2011), Detecting high contributions of primary organic matter to marine aerosol: A case study, Geophys. Res. Lett., 38, L02807, doi: 10.1029/2010GL046083.

Schmale, J., Schneider, J., Nemitz, E., Tang, Y. S., Dragosits, U., Blackall, T. D., Trathan, P. N., Phillips, G. J., Sutton, M., and Braban, C. F.: Sub-Antarctic marine aerosol: dominant contributions from biogenic sources, Atmos. Chem. Phys., 13, 8669-8694, https://doi.org/10.5194/acp-13-8669-2013, 2013.

Xu, L., Suresh, S., Guo, H., Weber, R. J., and Ng, N. L.: Aerosol characterization over the southeastern United States using high-resolution aerosol mass spectrometry: spatial and seasonal variation of aerosol composition and sources with a focus on organic nitrates, Atmos. Chem. Phys., 15, 7307-7336, https://doi.org/10.5194/acp-15-7307-2015, 2015.

---

## Author Comment (AC1) · 2 Oct 2018

**High spatial resolution mapping and source-apportionment of aerosol composition in Oakland, California using mobile aerosol mass spectrometery**

Responses by Shah et al. to referee comments

Referee comments are in regular font. Author responses are in red. Text quoted from the manuscript is *italicized*.

**Referee #1**

This manuscript shows high-resolution spatial patterns of PM composition and amount measured by an AMS on a truck in Oakland. Overall, I found the paper to be well within the scope of ACP, the analysis was well done, and the paper was well written. I have several comments that I'd like addressed before publication in ACP.

We thank the referee for their constructive feedback.

General comment (because it shows up in a few places):

Bottom  $\frac{1}{3}$  of P18, but also the abstract and point #3 on P23: To me, the simplest explanation as to why SV-OOA is higher in downtown is that its the location thats farthest downwind (wind typically heading from west to east in Figure S16). The mid-day gradient in SV-OOA appears to be steadily increasing from west to east in Figure 9, rather than a step change to higher values when entering downtown. The air starts picking up SOA precursors when it first hits land (either on the west or east sides of the bay) and SOA forms as the air moves across Oakland. Certainly more precursors are being added in Downtown, which helps, but the air as simply had more time to make SOA from the precursors (or oxidize the HOA and COA) by the time the air reaches downtown. The proposed hypotheses in the manuscript that deal with enhanced photochemical activity seem like less straightforward explanations. Downtown might be in a high-NOx regime, which would lower OH (acting against the proposed HONO source). Unless I'm missing something, I dont know why the simple amount of time spent over land hypothesis isnt prominent in the paper.

The referee raises a very good point in this paragraph: SV-OOA is steadily increasing with inland distance. This indeed suggests the simple time-over-land explanation for the observed SV-OOA trends in Figure 9 (Figure 8 in revised manuscript). We have added this discussion to the manuscript:

The increasing concentrations of SV-OOA with increasing inland distance (Figure 8) can be explained by the predominant Westerly winds (Figure S17). At the typical wind speed of ~ 15 kmph, downtown is 10-15 min downwind from the Port. The additional processing of OA in this time can result in higher SV-OOA in downtown.

However, this is not sufficient to explain the observed diurnal change in the spatial pattern of SV-OOA. The spatial gradient of SV-OOA is the largest in the morning (Figure 7 in revised manuscript), which is when primary emissions are also higher in downtown. This suggests that enhanced photochemistry could also be playing a role.

The referee mentions that downtown being in a high- $NO_x$  regime should lower the OH. This should be the case if the OH production rate is constant throughout the whole Oakland domain. However, if there is an additional OH source that helps to drive chemistry downtown (e.g., HONO emissions), then [OH] could be higher in that microenvironment. We did not measure VOC or  $NO_x$ , so we cannot directly address the question of LO-OOA production rates. However, as we discuss in the manuscript, the observation of elevated LO-OOA and sulfate downtown suggests faster photochemistry there. We offer HONO emissions from cars as one possible explanation. Herndon et al. (2008) previously showed that OOA formation scales with odd oxygen (NO2 + O3), and odd oxygen has been shown to be elevated in street canyons (Villena et al., 2011); this would be consistent with the additional LO-OOA we observe downtown.

Specific comments:

Abstract: The final sentence is subjective and unnecessary, in my opinion. The paper shows many things that would seem to be unique to Oakland (in addition to things that are likely common with other urban locations).

We removed this sentence.

P2 L14-15: The length and time scales dont seem to match. Winds would need to be very stagnant ( $\ll 1 \text{ km/hr}$ ) in order for "hundreds of meters" to correspond to "hours".

We reworded this sentence. It now reads: This rapid mixing occurs over tens to hundreds of meters downwind of the source (Canagaratna et al., 2010; Saha et al., 2018).

P3 L7: need a comma after "legislations"

We added a comma there.

P3 L10: I had to look up what "drayage" meant, and I suspect others might not know as well. Since the word is used quite a bit in the paper, it may be worth giving a very brief explanation here.

We added a definition of drayage trucks the first time the term is used. It now reads: Several prior studies have focused on air quality in Oakland because of the influence of ships and associated drayage trucks (trucks that transport cargo between the port and warehouses) driving through the residential district.

P6 L25-30: I know, generally, what bootstrapping statistical methods are, and I've used some. However, there was not enough info in this paragraph to really understand what was done (I could not repeat the analysis based on this description).

We added a description of the math. It now reads:

Bootstrap resampling: In order to compare observations of OA and its factors across areas influenced by different emissions (Port, West Oakland and downtown), we first determined the precision of these measurements by resampling the pool of data occurring in these areas. The strength (i.e., number of elements) of a bootstrapped dataset was the same as the strength of the dataset collected in that area. For instance: M = $\{m_1, m_2, ..., m_n\}$  was the original set of n measurements performed in an area (e.g., Port). From all n measurements in M, a measurement was randomly drawn to populate a synthetic set  $M'_1$ . Each random draw was from the original set M with replacement i.e., independent of previous draws. The synthetic set M' was populated until its strength was n (same as of the original set M). Generating such synthetic sets 104 times resulted in a bootstrapped pool  $M_B = \{M'_1, M'_2, \dots M'_{10^4}\}$ . From this, a bootstrap statistic set, S = $\{\tilde{s}_1, \tilde{s}_2, ... \tilde{s}_{10^4}\}$ , was created where  $\tilde{s}_i$  was the median of the synthetic set  $M_i$ . Finally, the difference between  $5^{th}$  and  $95^{th}$  percentile of all elements of S was used as a dispersion statistic of the median. This results in a bootstrapped median and its 95% confidence interval for the area whose data were chosen to be the original dataset M. We thus use the 95% confidence interval of the median as the precision of our measurements. Spatial differences larger than this precision are then deemed statistically significant. We are not

**Referee #2**

General comments:

This paper presents a detailed study on aerosol composition, focusing on the organic components, in Oakland, California. A mobile laboratory carrying an aerosol mass spectrometer (HR-ToF-AMS) was used to map the city roads in about 160 hours of measurements, resulting in a highly spatial resolved data set. The organic aerosol was separated using PMF into HOA, COA, and SV-OOA. While the manuscript is technically very sound and presents the AMS data analysis in great detail, it falls short in the interpretation of the results. To my opinion, more could have been done (or has to be done) in this direction, before publication in ACP can be recommended. My detailed comments are listed in the following.

We thank the referee for their constructive feedback.

**Major comments**

1. As written above, the interpretation of the results is not satisfying. That a large city like Oakland has a strong HOA influence would have been expected. The authors emphasize that Oakland has one of the largest shipping ports in the US. Thus, I would have expected to see as a result a separation of the HOA influence between ship exhausts, port-related traffic (trucks) and individual car traffic. I do not find any of this in the results or conclusions sections, only a little of such an analysis in section 3.2. The title suggests that not only the spatial distribution but also the sources of aerosol particles are investigated. It would have been interesting to learn about the reduction potential for the PM burden in a city like Oakland, e.g. whether reduction of ship emissions, truck or car traffic have a higher impact on PM loads.

We have made substantial changes to the section discussing BC measurements (Section 3.2 in ACPD version, section 3.4 in revised version with modified title). In addition to the OA/BC map that was in the ACPD version, we have added HOA/CO and BC/CO ratio maps to further bring out the influence of truck emissions at the Port. We also performed PMF separately on Port-only and downtown-only OA datasets and compared the two Port-HOA and downtown-HOA chemical mass spectra. However, unsurprisingly, the two factors look nearly identical, which underscores how challenging it is to determine HOA fraction from trucks v. cars. Regarding ship emissions: Tao et al. (2013) used particulate vanadium as a marker of ship emissions and found a median concentration of ~ 1 ng m-3 in West Oakland. We attempted to extract a vanadium signal in the AMS data (m/z 50.94), but the signal values were below the instrument detection limit (3 ng m-3).

Lastly, the term "sources" in our title simply refers to the chemical nature of OA and its apportionment to distinct emission/formation types e.g., traffic, cooking, etc. using PMF. We changed the title of our manuscript to clarify this.

2. The data analysis is restricted to AMS and black carbon data. But, as was written in section 2.2, also CO, CO2 and particle number concentration was measured. Apparently neither CO2 nor particle number were used here in the analysis. No correlation between CO and HOA is presented. I think that the addition of these parameters (e.g. ratios OA/CO, correlations between BC and CO, HOA and CO, HOA and particle number. . .) would be a benefit for the analysis.

We agree with the referee that we can make more from our measurements to distinguish emissions from cars and trucks e.g., by looking at relation between HOA and BC, HOA and CO, etc. We added a section addressing this truck/car comparison, as described in our response to the previous major comment.

Minor comments:

Page 2 line 24-25: Also in Paris (MEGAPOLI) COA was identified to be of high importance (Crippa et al., 2013b; Freutel et al., 2013)

We added this citation at this point in the text.

Page 7, Figure 2: The figure suffers from too much information. I suggest adding an extra figure in the Supplement with the number of unique samples and have in this figure only filled symbols color coded for OA.

Good suggestion. We followed it.

Page 8, lines 8.9: "Ambient measurements typically exhibit a positively-skewed distribution under the influence of local emission events." Really? Are three papers enough to say "typically"? And why is that so?

In ambient measurements, local emission events (e.g., driving by an open barbecue site while mobile sampling) and short-lived events (e.g., nucleation event measured by stationary monitor) will cause significantly high, narrow peaks in the time series of primary species (e.g., CO, NOx, particle number). The median is insensitive to these, but the mean can be very sensitive depending on number of data being averaged. Hence, mean > median and the distributions exhibit a positive skew. This behavior of pollutant probability distribution functions has been explained in Seinfeld and Pandis (2006). We cited the three papers because all of them performed mobile sampling in urban areas, and all of them attributed positive skews to local emissions. However, there are other examples, a mixture of stationary and mobile sampling studies, all of which report positively skewed distributions of primary species: Hoek et al. (2002); Apte et al. (2011); Von Der Weiden-Reinmüller et al. (2014); Tan et al. (2014, 2016); Zimmerman et al. (2018). We added the Seinfeld and Pandis textbook citation to our manuscript.

Page 10, lines 11-15: It was suggested by Sally Ngs group (e.g., Xu et al. (2015)) to replace "LV-OOA" and "SV-OOA" by "More-oxidized and less-oxidized oxygenated organic aerosol (MO-OOA and LO-OOA)". You may consider using LO-OOA instead of SV-OOA.

We replaced our factor terminology to LO- and MO-OOA.

Page 12: lines 0-5: Oceanic air masses: There is literature on marine OOA factors (e.g., Ovadnevaite et al., 2011; Schmale et al., 2013). There is some potential in this marine influence on aerosol properties. You could make more of it.

Schmale et al. (2013) reported a "MSA-OA (methanesulfonic acid OA)" factor. As the referee correctly pointed out, marine influence on aerosol properties can be important. By correlating particulate sulfate with this MSA mass contribution, Schmale et al. (2013) indeed show that marine influence on aerosol properties can be important (Figure 1A). However, we do not observe this marine influence in our data. Multiple lines of evidence point to this assessment: a) the correlation between particulate sulfate and MSA in Oakland (our data; Figure 1B) is  $R^2 = 0.12$ , while that of Schmale et al. (2013) is  $R^2 = 0.72$ , b) the ratio of MSA/sulfate in Oakland is only ~ 0.01, while that reported by Schmale et al. (2013) is ~ 0.25, c) the relative contribution of this MSA factor to the total OA in Oakland is less than 1% while that reported by Schmale et al. (2013) is 25%. That Oakland is an urban area and the measurement location of Schmale et al. (2013) was a remote island in the Antarctic explains these differences between the two datasets. Similarly, the work of Ovadnevaite et al. (2011) reported measurements in a remote location in Mace Head, Ireland which had a significant influence from marine OA with minor urban sources.

Figure 1: Correlation between particulate sulfate and methanesulfonic acid (MSA) OA reported in A) remote location of Bird Island research station in the sub-Antarctic region by Schmale et al. (2013) and B) Oakland (this study). Note: the axes on the two subplots are scaled differently. In the linear fit in the left subplot, Schmale et al. (2013) excluded data with M-OOA > 0.01  $\mu$ gm-3 (e.g., data cluster in the ellipse), where M-OOA was a highly oxygenated PMF factor (O/C > 1), attributed to background wind trajectories.

Contrasting to these two studies in remote locations, Crippa et al. (2013a) and Mohr et al. (2015) performed measurements in urban areas and their PMF results showed urban OA factors (HOA, COA, SV-OOA). Similar to our results from Oakland, Mohr et al. (2015) revealed no marine OA influence in Barcelona, despite it being a coastal location and receiving sea breezes. On the other hand, Crippa et al. (2013a) reported a marine factor in Paris. Paris receives influence from different directions (urban as well as clean marine wind masses). Due to these very different static contributions to the total OA, the PMF analysis of Crippa et al. (2013a) was able to identify a distinct marine factor. However, in the case of Oakland, we are unable to mathematically show a distinct marine factor presence because the wind directions remain relatively stable. As a result, rather than identify a distinct marine factor, PMF performs what is likely an artificial splitting, as explained in our discussion on "Quality of PMF solution".

We added the above figure and discussion to the supporting information.

Summary and conclusions: As already mentioned above, this section is very short. I think a statement on the relative influence of ships at the port, trucks at the port, and individual car traffic on the aerosol burden in the city would have been the desired output

of this study. We have added some text addressing this gap.

Technical Page 6, line 22: Link to appendix is missing We added this link.

Page 12, line 34: Oberdörster We corrected this error.

**Additional changes made by authors**

- All occurrences of  $\mu \text{gm}^{-3}$  were changed to  $\mu \text{g m}^{-3}$ .
- A few weeks after our submission to ACPD, the study of Preble et al. (2018) was published, reporting the exhaustion of diesel particulate filters and its influence on BC emissions at the Port of Oakland. This being very relevant to the conclusions drawn by us in this study, we have added a few sentences citing Preble et al. (2018) and highlighting the connection to our study (Sections 3.4 and 4 in revised version).

**References**

- Apte, J. S., Kirchstetter, T. W., Reich, A. H., Deshpande, S. J., Kaushik, G., Chel, A., Marshall, J. D. and Nazaroff, W. W. (2011), 'Concentrations of fine, ultrafine, and black carbon particles in auto-rickshaws in New Delhi, India', *Atmospheric Environment* 45(26), 4470–4480.
- Canagaratna, M. R., Onasch, T. B., Wood, E. C., Herndon, S. C., Jayne, J. T., Cross, E. S., Miake-Lye, R. C., Kolb, C. E. and Worsnop, D. R. (2010), 'Evolution of vehicle exhaust particles in the atmosphere', *Journal of the Air and Waste Management Association* **60**(10), 1192–1203.
- Crippa, M., Decarlo, P. F., Slowik, J. G., Mohr, C., Heringa, M. F., Chirico, R., Poulain, L., Freutel, F., Sciare, J., Cozic, J., Di Marco, C. F., Elsasser, M., Nicolas, J. B., Marchand, N., Abidi, E., Wiedensohler, A., Drewnick, F., Schneider, J., Borrmann, S., Nemitz, E., Zimmermann, R., Jaffrezo, J. L., Prévôt, A. S. and Baltensperger, U. (2013b), 'Wintertime aerosol chemical composition and source apportionment of the organic fraction in the metropolitan area of Paris', *Atmospheric Chemistry and Physics* 13(2), 961–981.
- Crippa, M., El Haddad, I., Slowik, J. G., Decarlo, P. F., Mohr, C., Heringa, M. F., Chirico, R., Marchand, N., Sciare, J., Baltensperger, U. and Prévôt, A. S. (2013a), 'Identification of marine and continental aerosol sources in Paris using high resolution aerosol mass spectrometry', *Journal of Geophysical Research Atmospheres* **118**(4), 1950–1963.
- Freutel, F., Schneider, J., Drewnick, F., Von Der Weiden-Reinmüller, S. L., Crippa, M., Prévôt, A. S., Baltensperger, U., Poulain, L., Wiedensohler, A., Sciare, J., Sarda-Estève, R., Burkhart, J. F., Eckhardt, S., Stohl, A., Gros, V., Colomb, A., Michoud, V., Doussin, J. F., Borbon, A., Haeffelin, M., Morille, Y., Beekmann, M. and Borrmann, S. (2013), 'Aerosol particle measurements at three stationary sites in the megacity of Paris during summer 2009: Meteorology and air mass origin dominate aerosol particle composition and size distribution', Atmospheric Chemistry and Physics 13(2), 933–959.
- Herndon, S. C., Onasch, T. B., Wood, E. C., Kroll, J. H., Canagaratna, M. R., Jayne, J. T., Zavala, M. A., Knighton, W. B., Mazzoleni, C., Dubey, M. K., Ulbrich, I. M., Jimenez, J. L., Seila, R., de Gouw, J. A., de Foy, B., Fast, J., Molina, L. T., Kolb, C. E. and Worsnop, D. R. (2008), 'Correlation of secondary organic aerosol with odd oxygen in Mexico City', *Geophysical Research Letters* 35(15), 1–6.
- Hoek, G., Meliefste, K., Cyrys, J., Lewné, M., Bellander, T., Brauer, M., Fischer, P., Gehring, U., Heinrich, J., Van Vliet, P. and Brunekreef, B. (2002), 'Spatial variability of fine particle concentrations in three European areas', *Atmospheric Environment* 36(25), 4077–4088.
- Mohr, C., DeCarlo, P. F., Heringa, M. F., Chirico, R., Richter, R., Crippa, M., Querol, X., Baltensperger, U. and Prevot, A. S. H. (2015), 'Spatial Variation of Aerosol Chemical Composition and Organic Components Identified by Positive Matrix Factorization in the Barcelona Region', *Environmental Science and Technology* 49(17), 10421–10430. URL: http://dx.doi.org/10.1021/acs.est.5b02149

- Ovadnevaite, J., O'Dowd, C., Dall'Osto, M., Ceburnis, D., Worsnop, D. R. and Berresheim, H. (2011), 'Detecting high contributions of primary organic matter to marine aerosol: A case study', *Geophysical Research Letters* 38(2), 2–6.
- Preble, C. V., Cados, T. E., Harley, R. A. and Kirchstetter, T. W. (2018), 'In-use performance and durability of particle filters on heavy-duty diesel trucks', *Environmental Science & Technology* 0(0), null. PMID: 30153019. URL: https://doi.org/10.1021/acs.est.8b02977
- Saha, P. K., Khlystov, A., Snyder, M. G. and Grieshop, A. P. (2018), 'Characterization of air pollutant concentrations, fleet emission factors, and dispersion near a North Carolina interstate freeway across two seasons', *Atmospheric Environment* 177(April 2017), 143–153.
  URL: https://doi.org/10.1016/j.atmosenv.2018.01.019
- Schmale, J., Schneider, J., Nemitz, E., Tang, Y. S., Dragosits, U., Blackall, T. D., Trathan, P. N., Phillips, G. J., Sutton, M. and Braban, C. F. (2013), 'Sub-Antarctic marine aerosol: Dominant contributions from biogenic sources', *Atmospheric Chemistry and Physics* 13(17), 8669–8694.
- Seinfeld, J. and Pandis, S. N. (2006), Atmospheric Chemistry and Physics: From Air Pollution to Climate Change, 2 edn, John Wiley and Sons, Inc., New York, chapter 21, Section 26.2.
- Tan, Y., Dallmann, T. R., Robinson, A. L. and Presto, A. A. (2016), 'Application of plume analysis to build land use regression models from mobile sampling to improve model transferability', Atmospheric Environment 134, 51–60. URL: http://dx.doi.org/10.1016/j.atmosenv.2016.03.032
- Tan, Y., Lipsky, E. M., Saleh, R., Robinson, A. L. and Presto, A. A. (2014), 'Characterizing the Spatial Variation of Air Pollutants and the Contributions of High Emitting Vehicles in Pittsburgh, PA.', *Environmental science & technology*. URL: http://dx.doi.org/10.1021/es5034074
- Tao, L., Fairley, D., Kleeman, M. J. and Harley, R. A. (2013), 'Effects of switching to lower sulfur marine fuel oil on air quality in the San Francisco Bay area', *Environmental Science and Technology* 47(18), 10171–10178.
  URL: http://dx.doi.org/10.1021/es401049x
- Villena, G., Kleffmann, J., Kurtenbach, R., Wiesen, P., Lissi, E., Rubio, M. A., Croxatto, G. and Rappenglück, B. (2011), 'Vertical gradients of HONO, NOx and O3 in Santiago de Chile', Atmospheric Environment 45(23), 3867–3873. URL: http://dx.doi.org/10.1016/j.atmosenv.2011.01.073
- Von Der Weiden-Reinmüller, S. L., Drewnick, F., Zhang, Q. J., Freutel, F., Beekmann, M. and Borrmann, S. (2014), 'Megacity emission plume characteristics in summer and winter investigated by mobile aerosol and trace gas measurements: The Paris metropolitan area', Atmospheric Chemistry and Physics 14(23), 12931–12950.
- Xu, L., Suresh, S., Guo, H., Weber, R. J. and Ng, N. L. (2015), 'Aerosol characterization over the southeastern United States using high-resolution aerosol mass spectrometry:

Spatial and seasonal variation of aerosol composition and sources with a focus on organic nitrates', Atmospheric Chemistry and Physics 15(13), 7307–7336.

Zimmerman, N., Li, H. Z., Ellis, A. A., Hauryliuk, A., Robinson, E. S., Gu, P., Shah, R. U., Ye, Q., Snell, L., R., S., Robinson, A. L., Apte, J. S. and Presto, A. A. (2018), 'Integrating spatiotemporal variability and modifiable factors into air pollution estimates: The center for air, climate, and energy solutions air quality observatory', Atmospheric Environment Submitted.

---

## Author Comment (AC2) · 2 Oct 2018

**High spatial resolution mapping and source-apportionment of aerosol composition  in Oakland, California using mobile aerosol mass spectrometry**

[revised manuscript text omitted]

**3.1 Organic aerosol (OA)**

We now discuss the spatial patterns of OA in more detail. Figure 3 compares the OA concentrations across the three areas (Port, West Oakland and downtown) using cumulative distribution function (CDF) curves in the upper panel. OA concentrations are spatially variable within each sampled area. A range of  > 2 $\mu$g m$^{-3}$ is observed in the median OA concentrations at
5    all magnets of each area.

The lower panel of Figure 3 shows the central tendency statistics (mean, median and standard deviation) of the values assigned to magnets in each area. The data are positively skewed in all polygons i.e., the mean is higher than the median. Ambient measurements typically exhibit a positively-skewed distribution under the influence of local emission events (Seinfeld and Pandis, 2006; Apte et al., 2017; Van den Bossche et al
10   Hence, the median is chosen over the mean as a central tendency statistic to discuss the OA spatial patterns. The results shown

in this figure are reinforced with statistical confidence by using bootstrap resampling (Figure S5). We determined a precision of 0.5 $\mu$g m$^{-3}$ in the median OA. Spatial differences larger than this precision are considered statistically significant.

[Figure]

**Figure 3.** Cumulative distributions, mean and median of OA concentrations in Port, West Oakland and downtown.

Downtown has a median OA concentration of 5.7 $\mu$g m$^{-3}$, which is 27% higher than West Oakland and Port. Almost the entire downtown CDF curve is $\sim$ 1.5 $\mu$g m$^{-3}$ greater than the West Oakland and Port curves, indicating that all parts of
5  downtown have higher OA concentrations than the rest of Oakland. Port and West Oakland have similar OA concentrations, as evidenced by their nearly-superimposed CDFs and similar medians ($4.6 \pm 0.05$ $\mu$g m$^{-3}$). However, Port measurements have more positive skewness than West Oakland; Port has a larger fraction of magnets with high OA concentrations than West Oakland. This suggests that the OA concentrations in Port are more influenced by local emission events. As mentioned earlier, Port has a high drayage truck activity, which likely explains this skewness. Results from bootstrapping support this explanation
10  (Figure S5). Port data have a higher mean (4.9 $\mu$g m$^{-3}$) with a wider 95% confidence interval (0.7 $\mu$g m$^{-3}$) about this mean, compared to West Oakland (mean: 4.6 $\mu$g m$^{-3}$; 95% confidence interval about the mean: 0.2 $\mu$g m$^{-3}$).

**3.2**

15

that while the vast majority (99%) of the truck fleet at the Port of Oakland is now equipped with advanced emission control technologies, the overall large volume of trucks arriving in the area results in higher BC emissions than light-duty gasoline vehicles.

We use the OA/BC ratio to distinguish car and truck emissions, with the underlying assumption that these use gasoline and diesel combustion, respectively. By extension, areas with smaller ambient OA/BC would indicate a larger influence from diesel truck emissions. Because OA and BC measurements were made at different sampling frequencies, we compare these measurements by first converting them to a synthetic 1 Hz timeframe as described earlier. A map of median OA/BC values at each 30 m magnet is shown in Figure **??**.

Fine-scale map of median OA/BC values (unitless).

In general, areas with lower OA/BC values are streets and highways approaching the Port of Oakland (e.g., Interstate 880 and the surface street connecting it to the port) as well as those around centers of shipping commerce in the Port. This is expected because trucks are found idling while in queue outside these facilities. Bootstrap resampling the BC dataset shows that the measurements (especially in Port) have a considerable positive skew. The ratio of mean to median BC in Port is higher (1.5) than both West Oakland and downtown (1.2). Because BC concentrations are largely influenced by local sources (diesel trucks), the large amount of drayage trucks in Port likely causes the mean BC to be ∼ 8% higher than downtown, even though median BC in downtown is ∼ 7% higher than Port. This finding suggests that despite substantial reductions in BC emissions from diesel trucks by the use of particulate filters and catalytic reducers, the influence of diesel truck emission plumes on the air quality in Port is higher than that in downtown.

**3.2 OA factors**

We identified three OA factors with distinct mass spectra using positive matrix factorization (PMF) of AMS data: hydrocarbon-like OA (HOA), cooking OA (COA) and semi-volatile less-oxidized oxygenated OA (SV-OOA LO-OOA). These factor profiles are shown in Figure 4. Distinct features of each factor mass spectra (e.g., signals at particular $m/z$'s and elemental O/C ratios) as well as the diurnal patterns in their time series are used to characterize these factors. We use the same nomenclature for these factors as has been used commonly in literature. For comparison, previously reported mass spectra of these factors are shown in Figure S9.

[Figure]

**Figure 4.** (a) Mass spectra, elemental ratios and average mass fraction of factors obtained from PMF analysis of all AMS spectra. (b) Boxplot of diurnal profiles of factors: rectangles enclose 1st through 3rd quartiles of data. Horizontal bars are medians. Asterisk markers are means. Whiskers are 5th and 95th percentiles.

*HOA:* The HOA factor has an elevated signal at the series of $C_nH_{2n+1}$ (e.g., $C_4H_9^+$ at $m/z$ 57) and $C_nH_{2n-1}$ (e.g., $C_3H_5^+$ at $m/z$ 41). Previous studies have identified this factor as a marker of fresh vehicular emissions based on its reduced state (O/C = 0.01) as well as its diurnal pattern (elevated during morning and evening traffic rush periods; Zhang et al., 2011; Mohr et al., 2012). HOA time series are highly positively skewed (average of hourly mean/median = 1.6) even during non-peak periods,
5  which indicates influence of local HOA plumes.

*COA:* The COA factor has a distinct signal at $m/z$ 55 ($C_4H_7^+$ and $C_3H_3O^+$). Previous studies have identified this factor as a marker of cooking emissions based on its reduced state (O/C = 0.11) as well as its diurnal pattern (elevated concentrations occurring typically during lunch and dinner time; Mohr et al., 2009, 2012; Zhang et al., 2011). Similar to HOA, COA time series are also highly positively skewed (average of hourly mean/median = 1.53), indicating influence of local COA plumes.

10  *LO-OOA:* Compared to the HOA and COA factors, this factor is relatively more oxygenated with a distinct peak at $m/z$ 44 in its mass spectrum. Secondary OA contains oxygen-containing groups (e.g., carboxylic acids, alcohols and carbonyls). These groups, upon ionization in the AMS, contribute to the $m/z$ 44 ($COO^+$) signal. Generally, based on increasing extent of atmospheric processing two classes of oxygenated OA are identified by signals at $m/z$ 43 and 44: less-
15  and more-oxidized oxygenated OA (LO-OOA and MO-OOA, respectively; Xu et al., 2015). Of these two, LO-OOA is relatively less oxygenated, bears similarity to semi-volatile OOA (SV-OOA) in the two-dimensional volatility basis set

[revised manuscript text omitted]

 The increasing concentrations of LO-OOA with increasing inland distance (Figure 8) can be explained by the predominant Westerly winds (Figure S17). At the typical wind speed of $\sim 15$ kmph, downtown is 10-15 min downwind from the Port. The additional processing of OA in this time can result in higher LO-OOA in downtown. However, we also observe that the spatial pattern of LO-OOA changes diurnally, with a stronger spatial gradient in the morning than the afternoon (Figure 7). This suggests that the local photochemical microenvironment may also contribute to the observed LO-OOA spatial pattern.

Enhanced photochemical production of SOA could be due to several reasons. Firstly, the pool of reactive SOA precursor vapors is likely enhanced in downtown relative to other areas. We show above in Figure 7 that HOA is highest in downtown. HOA is co-emitted with volatile and intermediate-volatility organic compounds that efficiently generate SOA upon photochemical oxidation (Robinson et al., 2007; Gordon et al., 2013; Zhao et al., 2015).

Secondly, concentrations of the hydroxyl (OH) radical may also be higher downtown. The OH radical is the dominant daytime oxidizing agent, especially for reduced compounds emitted from motor vehicles. It has been shown that in urban, polluted environments (street canyons), high-$NO_x$ emissions from vehicles render increased levels of nitrous acid (HONO), which in turn results in increased local oxidation capacity (Yun et al., 2017; Villena et al., 2011) due to rapid photolysis of HONO to OH (Stutz et al., 2000; Finlayson-Pitts and Pitts, 2000; Zhong et al., 2017).

A third factor may also contribute to higher precursor concentrations, and therefore additional LO-OOA, in downtown. The presence of tall buildings in downtown can create higher surface roughness, which in turn can reduce pollutant dispersion and promote internal recirculation (Zhong et al., 2017). This possibility is further examined in Figure S19, which shows building heights in downtown. Downtown has relatively taller buildings that collectively act as a wall parallel to the Interstate 980 highway upwind. The air masses in downtown may experience stagnation and poor ventilation relative to West Oakland and Port. This, in turn, can increase the reaction time, thereby allowing more local  LO-OOA formation.

The combined impact of vehicle emissions on OH (via HONO) and gas-phase precursor concentrations in downtown would be expected to be largest in the morning, since these are co-emitted with HOA. Indeed, the largest enhancement of  LO-OOA in downtown occurs in the morning hours, at the same time as the largest enhancement of HOA. From morning to mid-day,  LO-OOA concentrations become more spatially uniform; median  LO-OOA concentrations in West Oakland and Port increase by 23% (0.6 $\mu$g m$^{-3}$) and 16% (0.5 $\mu$g m$^{-3}$) respectively, while downtown only increases by 7% (0.25 $\mu$g m$^{-3}$). This suggests that while downtown has high photochemical activity in the morning, the entire sampling domain transitions towards a more uniform photochemical state by mid-day.

We further investigated the enhanced photochemical activity in downtown by analyzing mobile measurements of particulate sulfate ($SO_4^{2-}$). $SO_4^{2-}$ is formed upon reaction of gas-phase $SO_2$ with OH (Miyakawa et al., 2007), which can be enhanced in polluted environments due to catalytic involvement of black carbon particles (Novakov et al., 1974). Figure S8 shows that $SO_4^{2-}$ concentrations in downtown are higher ($\sim 8\%$) than Port and West Oakland, a trend similar to that observed in  LO-OOA concentrations.

Ships associated with the Port are the major source of $SO_2$ in Oakland (Tao et al., 2013). While ship $SO_2$ emissions have significantly decreased in recent years, ships remain the dominant $SO_2$ source in our sampling domain, and there seem to be no local sources of $SO_2$ or particulate $SO_4^{2-}$ in downtown. Thus, elevated concentrations of secondary sulfate in downtown would therefore support the hypothesis that OH concentrations are also higher in downtown, giving rise to local variations in $SO_4^{2-}$ formation rate.

**3.4  Spatial patterns of gasoline and diesel vehicle emissions**

In this subsection, we compare the influence of emissions from diesel trucks against that from gasoline-powered vehicles. We do this by comparing concentrations and ratios of OA, HOA, BC, and CO. Traditionally, diesel vehicles have significantly higher emissions of HOA and BC than gasoline vehicles (Ban-Weiss et al., 2008), whereas gasoline vehicles have higher CO emissions than diesel vehicles (May et al., 2014). Vehicle emission standards are regularly tightened, so emissions are lower for newer vehicles. However, emission reductions are not equal for all species, so the ratios of different emitted pollutants vary with both vehicle age and fuel type. Thus, the specific emissions and emission ratios from the vehicle fleet in a city, or a portion of the city, depend on both the gasoline-diesel split and the age distribution of gasoline and diesel vehicles.

Based on typical emissions from gasoline and diesel vehicles, we would expect that: (a) all areas with heavy traffic should have elevated concentrations OA, HOA, BC, and CO, and (b) diesel-dominated areas will have higher BC/CO, higher HOA/CO, and lower OA/BC ratios than gasoline-dominated or background locations. The specific concentration ratios in different areas

will be a function of the vehicle fleet in that area and the contribution of fresh emissions versus background. One potential drawback of using concentration ratios to identify gasoline- versus diesel-dominated areas is that emission ratios from cars are becoming more "diesel-like" (May et al., 2014; Saliba et al., 2017), which complicates the analysis for newer vehicle fleets.

[Figure]

**Figure 9.** Fine-scale maps of (a) OA/BC (unitless), (b) BC/CO and (c) HOA/CO. Subfigure (d) shows similarity between the HOA factors identified by factorization of Port-only and downtown-only OA data. The two spectra are dominated by hydrocarbons ($C_xH_y$) and are highly similar to each other.

Figure 9A-C shows spatially-resolved OA/BC, BC/CO, and HOA/BC ratios. Because measurements were made at different sampling frequencies, we generate the ratios by first converting the measurements to a synthetic 1 Hz time base as described earlier. All three ratios show an influence of diesel trucks (lower OA/BC, higher BC/CO and HOA/BC) at the Port, on Interstates 880 and 980, and on truck routes that connect I-880 to the Port. The absolute concentrations of BC are also elevated in these areas (Figure S20). Bootstrap resampling the BC dataset shows that the measurements (especially in Port) have a considerable positive skew (Figure S21). The ratio of mean to median BC in Port is higher (1.5) than both West Oakland and downtown (1.2). Because BC concentrations are largely influenced by local sources (diesel trucks), the large amount of drayage trucks in Port likely causes the mean BC to be $\sim 8\%$ higher than downtown, even though median BC in downtown is $\sim 7\%$ higher than Port.

All three ratios suggest less diesel influence in West Oakland and downtown than in Port. BC/CO ratios in the Port are generally 6-8 ng m$^{-3}$ ppb$^{-1}$, whereas much of West Oakland and downtown have BC/CO $\sim 4$ ng m$^{-3}$ ppb$^{-1}$. Some hotspots of BC/CO appear in industrial parts of West Oakland, including (a) an area near a metals recycling facility in West Oakland highlighted by Apte et al. (2017) as a location with high diesel traffic and (b) a part of Grand Avenue which is also a truck route. A hotspot of BC/CO also appears on, and downwind of, I-980 in downtown.

The HOA/CO ratio paints a similar, but not identical picture as the BC/CO ratio. West Oakland and downtown have lower HOA/CO than the Port, suggesting less diesel truck influence. However, downtown has higher HOA/CO than West Oakland. Since gasoline cars appear to be the major traffic source in West Oakland and downtown, we would expect a similar HOA/CO ratio in these areas. However, several factors may contribute to the higher HOA/CO in downtown: (a) the downtown and West Oakland fleets are likely not identical (e.g., more diesel buses in downtown), (b) differences in driving mode, with more stop-and-go driving in downtown than West Oakland, and (c) larger contribution of background CO to the total measured CO in West Oakland.

The overall picture painted by Figure 9A-C is that the Port is more impacted by diesel vehicle emissions than downtown and West Oakland. Downtown has high HOA (Figure S18) because of high traffic volumes, but this area seems to be dominated by gasoline vehicles with a smaller diesel contribution. The large diesel influence at the Port persists in spite of recent, aggressive efforts to upgrade the drayage truck fleet so that the majority (99%) of drayage trucks now have diesel particulate filters (Preble et al., 2015). As shown by Dallmann et al. (2011) and Preble et al. (2015), overall BC emissions from the Port truck fleet fell by $\sim 75\%$ between 2010 and 2013, suggesting that HOA and BC concentrations at the Port were higher in the past. Further reductions in BC and HOA at the Port could be achieved by addressing high emitters; measurements by Preble et al. (2018) in 2015 showed that 7% of the drayage truck fleet at the Port accounted for 65% of the total emitted BC because the diesel particulate filters on these trucks were failing.

Since the vehicle fleet appears to be significantly different between Downtown and Port, we attempted to derive separate HOA factors for these two areas as a means to directly quantify gasoline versus diesel emissions using the AMS. We isolated Port OA data from downtown OA data and factorized them separately using PMF. The HOA factors identified for Port and downtown nearly identical ($R^2 = 0.97$; Figure 9D). This is likely due to a combination of similar emissions (HOA from gasoline and diesel

excessive fragmentation upon ionization in the AMS. The similarity in the Port and Downtown HOA factors makes it essentially impossible to distinguish HOA emitted by diesel trucks from HOA emitted by gasoline cars.

**3.5 Elemental analysis of bulk and factor OA**

[Figure]

[revised manuscript text omitted]
., 2015).  Further, more than 70% of the  ships at the Port now utilize shore power provision and thus do not idle their engines. It is thus reasonable to expect that the influence of ship emissions on the air quality in Oakland  has substantially reduced after the measurements of Tao et al. (2013). Enforced installation of diesel particulate filters on drayage trucks has also significantly reduced truck emissions at the Port Dallmann et al. (2011); Preble et al. (2015), although a recent finding by Preble et al. (2018) has raised concern about the exhaustion of these filters and the resultant increase in truck emissions.

[revised manuscript text omitted]

Saliba, G., Saleh, R., Zhao, Y., Presto, A. A., Lambe, A. T., Frodin, B., Sardar, S., Maldonado, H., Maddox, C., May, A. A., Drozd, G. T., Goldstein, A. H., Russell, L. M., Hagen, F., and Robinson, A. L.: Comparison of Gasoline Direct-Injection (GDI) and Port Fuel Injection

(PFI) Vehicle Emissions: Emission Certification Standards, Cold-Start, Secondary Organic Aerosol Formation Potential, and Potential Climate Impacts, Environmental Science and Technology, 51, 6542–6552, https://doi.org/10.1021/acs.est.6b06509, 2017.

Schmale, J., Schneider, J., Nemitz, E., Tang, Y. S., Dragosits, U., Blackall, T. D., Trathan, P. N., Phillips, G. J., Sutton, M., and Braban, C. F.: Sub-Antarctic marine aerosol: Dominant contributions from biogenic sources, Atmospheric Chemistry and Physics, 13, 8669–8694, https://doi.org/10.5194/acp-13-8669-2013, 2013.

Seinfeld, J. and Pandis, S. N.: Atmospheric Chemistry and Physics: From Air Pollution to Climate Change, chap. 21, Section 26.2, John Wiley and Sons, Inc., New York, 2 edn., 2006.

Stölzel, M., Breitner, S., Cyrys, J., Pitz, M., Wölke, G., Kreyling, W., Heinrich, J., Wichmann, H. E., and Peters, A.: Daily mortality and particulate matter in different size classes in Erfurt, Germany, Journal of Exposure Science and Environmental Epidemiology, 17, 458–467, https://doi.org/10.1038/sj.jes.7500538, http://dx.doi.org/10.1038/sj.jes.7500538, 2007.

Stutz, J., Kim, E. S., Platt, U., Bruno, P., Perrino, C., and Febo, A.: UV-visible absorption cross sections of nitrous acid, Journal of Geophysical Research Atmospheres, 105, 14 585–14 592, https://doi.org/10.1029/2000JD900003, https://www.scopus.com/inward/record.uri?eid=2-s2.0-0033793746{&}partnerID=40{&}md5=df56e17158e0b457ca4fe8693f776846, 2000.

Sueper, D., Allan, J. D., Dunlea, E., Crosier, J., Kimmel, J. R., DeCarlo, P. F., Aiken, A. C., and Jimenez, J. L.: A Community Software for Quality Control and Analysis of Data from the Aerodyne Time-of-Flight Aerosol Mass Spectrometers (ToF-AMS), Tech. rep., Reno, NV, 2007.

Sun, Y. L., Zhang, Q., Schwab, J. J., Demerjian, K. L., Chen, W. N., Bae, M. S., Hung, H. M., Hogrefe, O., Frank, B., Rattigan, O. V., and Lin, Y. C.: Characterization of the sources and processes of organic and inorganic aerosols in New York city with a high-resolution time-of-flight aerosol mass apectrometer, Atmospheric Chemistry and Physics, 11, 1581–1602, https://doi.org/10.5194/acp-11-1581-2011, http://dx.doi.org/10.5194/acp-11-1581-2011, 2011.

Tao, L., Fairley, D., Kleeman, M. J., and Harley, R. A.: Effects of switching to lower sulfur marine fuel oil on air quality in the San Francisco Bay area, Environmental Science and Technology, 47, 10 171–10 178, https://doi.org/10.1021/es401049x, http://dx.doi.org/10.1021/es401049x, 2013.

Tsigaridis, K., Krol, M., Dentener, F. J., Balkanski, Y., Lathière, J., Metzger, S., Hauglustaine, D. A., and Kanakidou, M.: Change in global aerosol composition since preindustrial times, Atmospheric Chemistry and Physics, 6, 5143–5162, https://doi.org/10.5194/acp-6-5143-2006, http://dx.doi.org/10.5194/acp-6-5143-2006, 2006.

Ulbrich, I. M., Canagaratna, M. R., Zhang, Q., Worsnop, D. R., and Jimenez, J. L.: Interpretation of Organic Components from Positive Matrix Factorization of Aerosol Mass Spectrometric Data., Atmos. Chem. Phys., 9, 2891, https://doi.org/10.5194/acp-9-2891-2009, http://dx.doi.org/10.5194/acp-9-2891-2009, 2009.

Ulbrich, I. M., Handschy, A., Lechner, M., and Jimenez, J. L.: High-Resolution AMS Spectral Database, http://cires1.colorado.edu/jimenez-group/HRAMSsd/, 2018.

United Nations: World Urbanization Prospects, Tech. rep., United Nations, Department of Economic and Social Affairs, https://doi.org/10.4054/DemRes.2005.12.9, http://dx.doi.org/10.4054/DemRes.2005.12.9, 2014.

US Census: US Census Bureau Reporter - Oakland, CA, https://censusreporter.org/profiles/16000US0653000-oakland-ca/, 2016.

Van den Bossche, J., Peters, J., Verwaeren, J., Botteldooren, D., Theunis, J., and De Baets, B.: Mobile monitoring for mapping spatial variation in urban air quality: Development and validation of a methodology based on an extensive dataset, Atmospheric Environment, 105, 148–161, https://doi.org/10.1016/j.atmosenv.2015.01.017, http://dx.doi.org/10.1016/j.atmosenv.2015.01.017, 2015.

Villena, G., Kleffmann, J., Kurtenbach, R., Wiesen, P., Lissi, E., Rubio, M. A., Croxatto, G., and Rappenglück, B.: Vertical gradients of HONO, NOx and O3 in Santiago de Chile, Atmospheric Environment, 45, 3867–3873, https://doi.org/10.1016/j.atmosenv.2011.01.073, http://dx.doi.org/10.1016/j.atmosenv.2011.01.073, 2011.

Von Der Weiden-Reinmüller, S. L., Drewnick, F., Zhang, Q. J., Freutel, F., Beekmann, M., and Borrmann, S.: Megacity emission plume characteristics in summer and winter investigated by mobile aerosol and trace gas measurements: The Paris metropolitan area, Atmospheric Chemistry and Physics, 14, 12 931–12 950, https://doi.org/10.5194/acp-14-12931-2014, , 2014.

Worton, D. R., Isaacman, G., Gentner, D. R., Dallmann, T. R., Chan, A. W., Ruehl, C., Kirchstetter, T. W., Wilson, K. R., Harley, R. A., and Goldstein, A. H.: Lubricating oil dominates primary organic aerosol emissions from motor vehicles, Environmental Science and Technology, 48, 3698–3706, https://doi.org/10.1021/es405375j, 2014.

Xu, L., Suresh, S., Guo, H., Weber, R. J., and Ng, N. L.: Aerosol characterization over the southeastern United States using high-resolution aerosol mass spectrometry: Spatial and seasonal variation of aerosol composition and sources with a focus on organic nitrates, Atmospheric Chemistry and Physics, 15, 7307–7336, https://doi.org/10.5194/acp-15-7307-2015, 2015.

Ye, Q., Gu, P., Li, H. Z., Robinson, E. S., Lipsky, E., Kaltsonoudis, C., Lee, A. K., Apte, J. S., Robinson, A. L., Sullivan, R. C., Presto, A. A., and Donahue, N. M.: Spatial Variability of Sources and Mixing State of Atmospheric Particles in a Metropolitan Area, Environmental Science & Technology, 0, null, https://doi.org/10.1021/acs.est.8b01011, https://doi.org/10.1021/acs.est.8b01011, pMID: 29775536, 2018.

Yuan, C., Ng, E., and Norford, L. K.: Improving air quality in high-density cities by understanding the relationship between air pollutant dispersion and urban morphologies, Building and Environment, 71, 245–258, https://doi.org/10.1016/j.buildenv.2013.10.008, http://dx.doi.org/10.1016/j.buildenv.2013.10.008, 2014.

Yun, H., Wang, Z., Zha, Q., Wang, W., Xue, L., Zhang, L., Li, Q., Cui, L., Lee, S., Poon, S. C., and Wang, T.: Nitrous acid in a street canyon environment: Sources and contributions to local oxidation capacity, Atmospheric Environment, 167, 223–234, https://doi.org/10.1016/j.atmosenv.2017.08.018, http://dx.doi.org/10.1016/j.atmosenv.2017.08.018, 2017.

Zhang, Q., Jimenez, J. L., Canagaratna, M. R., Allan, J. D., Coe, H., Ulbrich, I., Alfarra, M. R., Takami, A., Middlebrook, A. M., Sun, Y. L., Dzepina, K., Dunlea, E., Docherty, K., DeCarlo, P. F., Salcedo, D., Onasch, T., Jayne, J. T., Miyoshi, T., Shimono, A., Hatakeyama, S., Takegawa, N., Kondo, Y., Schneider, J., Drewnick, F., Borrmann, S., Weimer, S., Demerjian, K., Williams, P., Bower, K., Bahreini, R., Cottrell, L., Griffin, R. J., Rautiainen, J., Sun, J. Y., Zhang, Y. M., and Worsnop, D. R.: Ubiquity and dominance of oxygenated species in organic aerosols in anthropogenically-influenced Northern Hemisphere midlatitudes, Geophysical Research Letters, 34, 1–6, https://doi.org/10.1029/2007GL029979, http://dx.doi.org/10.1029/2007GL029979, 2007.

Zhang, Q., Jimenez, J. L., Canagaratna, M. R., Ulbrich, I. M., Ng, N. L., Worsnop, D. R., and Sun, Y.: Understanding atmospheric organic aerosols via factor analysis of aerosol mass spectrometry: A review, Analytical and Bioanalytical Chemistry, 401, 3045–3067, https://doi.org/10.1007/s00216-011-5355-y, http://dx.doi.org/10.1007/s00216-011-5355-y, 2011.

Zhao, Y., Nguyen, N. T., Presto, A. A., Hennigan, C. J., May, A. A., and Robinson, A. L.: Intermediate Volatility Organic Compound Emissions from On-Road Diesel Vehicles: Chemical Composition, Emission Factors, and Estimated Secondary Organic Aerosol Production, Environmental Science and Technology, 49, 11 516–11 526, https://doi.org/10.1021/acs.est.5b02841, 2015.

Zhong, J., Cai, X. M., and Bloss, W. J.: Large eddy simulation of reactive pollutants in a deep urban street canyon: Coupling dynamics with O3-NOx-VOC chemistry, Environmental Pollution, 224, 171–184, https://doi.org/10.1016/j.envpol.2017.01.076, http://dx.doi.org/10.1016/j.envpol.2017.01.076, 2017.

Zimmerman, N., Li, H. Z., Ellis, A. A., Hauryliuk, A., Robinson, E. S., Gu, P., Shah, R. U., Ye, Q., Snell, L., R., S., Robinson, A. L., Apte, J. S., and Presto, A. A.: Integrating Spatiotemporal Variability and Modifiable Factors into Air Pollution Estimates: The Center for Air, Climate, and Energy Solutions Air Quality Observatory, Atmospheric Environment, Submitted, 2018.

*Supplemental information for*

**High spatial resolution mapping of aerosol composition and sources in Oakland, California using mobile aerosol mass spectrometry**

Shah et al.

*Correspondence to:* Albert A. Presto (apresto@andrew.cmu.edu)

**A1 Accounting for temporal trends**

Over the course of mobile sampling, the urban background air quality can have daily and diurnal variations due to meteorological changes. These variations can be accounted for with the help of concurrent stationary measurements performed at an urban background location, provided this background location is not in close proximity ($\leq$ 50 m) of a major emission source (e.g., a street with > 3000 daily vehicles, construction sites, industrial emissions, etc; Hoek et al., 2002; Van den Bossche et al., 2015). Further, different meteorological conditions (temperature, relative humidity, solar irradiation) influence individual components of particulate matter differently (e.g., the gas-particle partitioning of $NO_3^-$, $SO_4^{2-}$ and OA have different sensitivities to temperature; more irradiation can indirectly result in more secondary OA). Ideally, the temporal correction of mobile measurements of a particular PM component would be performed using background measurements of that individual component performed in the same manner. Thus, mobile AMS measurements should ideally be corrected using concurrent stationary AMS measurements (Mohr et al., 2015).

We did not perform concurrent stationary AMS measurements in Oakland in this campaign. Stationary measurements of criteria pollutants (CO, $NO_x$, $PM_{2.5}$, etc.) were made by a regulatory monitor operated by the Bay Area air quality management district. However, this monitor was located in a parking lot within $\sim$ 30 m of a major street (> 20,000 vehicles daily; Knoderer et al., 2016). Due to this, we did not use this monitor as an indicator of diurnal variations in urban background. Instead, we assumed that by shuffling the order of visiting each polygon within the sampling domain, repeated measurements at a given point in space were well-balanced in time. We validated this assumption by running the timestamps of our data through the spatial aggregation routine described earlier. The resulting map (Figure S1) confirms that the measurements in this study were indeed well-balanced in space and time. Further, diurnal variations in meteorological factors had a consistent pattern on all days of this campaign (Figure S2), hence by balancing our samples in space and time, we assume that the effects of diurnal variations in urban background are nullified upon aggregation of data from multiple days.

Figure S2 shows that daily variations in meteorological factors is typically larger and sporadic relative to diurnal variations. To account for the possible effect of these daily variations on urban background, we tested a daily multiplicative correction factor to our mobile measurements based on stationary measurements of $PM_{2.5}$ at the regulatory site.

[Figure]

**Figure S1.** Spatially aggregated timestamp of all AMS measurements. Typically, drives were performed from 0800 to 1800 hours. If all parts of the domain were sampled randomly everyday, the average of all measurement timestamps should ideally occur at 1300 hours. This figure shows that this ideality was indeed realized in this campaign and that we balanced our measurements in space and time reasonably well. On most days, we used the south- and north-bound lanes of Interstate 980 to enter and exit the domain at 0800 and 1800 hours, respectively. As a result, there is a temporal bias in sampling this highway, as shown by the figure.

[Figure]

**Figure S2.** Diurnal and daily variations in meteorological factors in Oakland during the sampling period. Diurnal variations have a consistent pattern that repeats everyday. By comparison, daily variations have more sporadic variations.

[Figure]

**Figure S3.** Effect of applying a daily correction factor to mobile measurements of a) $PM_1$ and b) OA. The 95% confidence intervals are achieved from bootstrapping and represent the difference between the $5^{th}$ and $95^{th}$ percentiles of the bootstrapped medians. Each day's correction factor is calculated as $CF_i = \dfrac{C_i}{C_{campaign}}$, where $C_i$ is the median $PM_{2.5}$ concentration measured on the $i^{th}$ day of the campaign, and $C_{campaign}$ is the median $PM_{2.5}$ concentration measured on all days of the campaign. Stationary $PM_{2.5}$ measurements were made by the regulatory monitor in West Oakland (Knoderer et al., 2016). Since we typically drove everyday from 8 AM to 6 PM, we only used stationary data from this daily time period.

[Figure]

**Figure S4.** Sampling coverage of the domain. Inset: Cumulative distribution of raw and unique samples in all magnets in domain.

**A3 Bootstrapping analysis**

[Figure]

**Figure S5.** Polygon-specific cumulative distribution function (CDF) curves of spatially-aggregated OA concentrations. *Insets:* Probability distribution function (PDF) histograms for central tendency statistics (mean and median) of synthetic datasets created using bootstrap resampling. The PDF histogram for bootstrapped medians is shown with a coarser bin-width to guide the eye better. The abscissae on the insets have the same units as the parent abscissa, but with a zoomed-in scale.

~~Polygon-specific cumulative distribution function (CDF) curves of black carbon (BC) concentrations. *Insets:* Probability distribution function (PDF) histograms for central tendency statistics (mean and median) of synthetic datasets created using bootstrap resampling of raw data. The PDF histograms are shown with a coarser bin-width to guide the eye better. The abscissae on the insets have the same units as the parent abscissa, but with a zoomed-in scale.~~

[Figure]

**Figure S6.** Cumulative distribution functions of primary fraction of OA (i.e., COA + HOA), resolved by area. Insets are results of bootstrapping.

[Figure]

**Figure S7.** Bootstrapped statistics of OA and its factors, resolved by area and by time periods.

[Figure]

**Figure S8. A:** Median $SO_4^{2-}$ at all magnets. **B:** Cumulative distribution function (CDF) curves of spatially-aggregated $SO_4^{2-}$ concentrations. **C:** CDF curves of $SO_4^{2-}$ concentrations without spatial aggregation. *Insets:* Probability distribution function (PDF) histograms for central tendency statistics (mean and median) of synthetic datasets created using bootstrap resampling of raw data. The abscissae on the insets have the same units as the parent abscissa, but with a zoomed-in scale.

It is seen that the CDFs of all polygons become "tighter" after spatial aggregation, which is expected, since the spatial aggregation routine reduces the spread in measurements via data reduction. However, the evidence of downtown being having a higher median (and especially higher mean) $SO_4^{2-}$ than Port and West Oakland is not diminished by this data reduction, confirming that this trend is consistent across all days of sampling. Further, the $SO_4^{2-}$ data exhibit minor positive skewness, suggesting that there are no local point sources of $SO_4^{2-}$ in the domain.

**A4 Quality of PMF solution**

[Figure]

**Figure S9.** PMF factor mass spectra identified in **A.** Oakland and **B.** Barcelona (Mohr et al., 2015). Barcelona spectra obtained from the online high-resolution spectral database (Ulbrich et al., 2018, 2009).

[Figure]

**Figure S10.**  LO-OOA factor time series matched against $NO_3^-$, a marker for semi-volatile photochemically aged OA.

[Figure]

**Figure S11.** Residuals of three-factor PMF solution. The boxplots show the median (centerline) and quartiles (box limits) of the scaled residuals.

[Figure]

**Figure S12.** Mass spectra and elemental ratios of the four factors obtained from PMF analysis

[Figure]

**Figure S13.** Residuals of four-factor PMF solution. The boxplots show the median (centerline) and quartiles (box limits) of the scaled residuals.

[Figure]

**Figure S14.** Variation in $Q/Q_{exp}$ and mass fractions of factors with $f_{peak}$ for 3- and 4-factor solutions. $Q$ = sum of square of scaled residuals. $Q_{exp}$ = number of degrees of freedom of the fitted data. $Q/Q_{exp}$ should be $\approx$ 1. $f_{peak}$ is a rotational parameter that can be chosen to examine different possible variations within an $n$-factor solution (Ulbrich et al., 2009).

[Figure]

**Figure S15.** The Van Krevelen plane. Gray points represent all OA measurements in this study. Diamonds represent OA factors identified in the 3-factor (**A.**) and 4-factor (**B.**) PMF solution. Different oxygenation pathways are shown by the black dotted lines. Dotted blue lines are isopleths for average carbon oxidation states ($\overline{OS}_C$ = 2×O/C − H/C). The region of ambient oxygenated OA measurements, as reported by Ng et al. (2011) is shown between the dashed curves.

**Marine influence.**

Marine influence on aerosol properties has been shown to be important in some locations. For instance, Schmale et al. (2013) reported a "MSA-OA (methanesulfonic acid OA)" factor in their study on a remote island in the Antarctic ocean. By correlating particulate sulfate with this MSA mass contribution, Schmale et al. (2013) indeed show that marine influence on aerosol properties can be important (Figure S16A). We do not observe this marine influence in our data. Multiple lines of evidence point to this assessment: a) the correlation between particulate sulfate and MSA in Oakland (our data; Figure S16B) is $R^2 = 0.12$, while that of Schmale et al. (2013) is $R^2 = 0.72$, b) the ratio of MSA/sulfate in Oakland is only $\sim 0.01$, while that reported by Schmale et al. (2013) is $\sim 0.25$, c) the relative contribution of this MSA factor to the total OA in Oakland is less than 1% while that reported by Schmale et al. (2013) is 25%. That Oakland is an urban area and the measurement location of Schmale et al. (2013) was a remote island in the Antarctic explains these differences between the two datasets. Similarly, the work of Ovadnevaite et al. (2011) reported measurements in a remote location in Mace Head, Ireland which had a significant influence from marine OA with minor urban sources.

[Figure]

**Figure S16.** Correlation between particulate sulfate and methanesulfonic acid (MSA) OA reported in A) remote location of Bird Island research station in the sub-Antarctic region by Schmale et al. (2013) and B) Oakland (this study). Note: the axes on the two subplots are scaled differently. In the linear fit in the left subplot, Schmale et al. (2013) excluded data with M-OOA > 0.01 $\mu$gm$^{-3}$ (e.g., data cluster in the ellipse), where M-OOA was a highly oxygenated PMF factor (O/C > 1), attributed to background wind trajectories.

Contrasting to these two studies in remote locations, Crippa et al. (2013a) and Mohr et al. (2015) performed measurements in urban areas and their PMF results showed urban OA factors (HOA, COA, SV-OOA). Similar to our results from Oakland, Mohr et al. (2015) revealed no marine OA influence in Barcelona, despite it being a coastal location and receiving sea breezes. On the other hand, Crippa et al. (2013a) reported a marine factor in Paris. Paris receives influence from different directions (urban as well as clean marine wind masses). Due to these very different static contributions to the total OA, the PMF analysis of Crippa et al. (2013a) was able to identify a distinct marine factor. However, in the case of Oakland, we are unable to mathematically show a distinct marine factor presence because the wind directions remain relatively stable. As a result, rather

than identify a distinct marine factor, PMF performs what is likely an artificial splitting, as explained previously in this section.

**A5 Wind measurements**

[Figure]

**Figure S17.** A map of the San Francisco (SF) Bay area, showing the location of the sampling domain in Oakland relative to SF city and downtown. Wind rose plots show hourly median wind speed and direction measured at two stationary anemometers in Oakland and San Francisco (BAAQMD, 2018). Purple wind rose arrows are wind measurements acquired during periods of concurrent mobile sampling (typically 8 am to 6 pm), while black arrows are measurements from when mobile sampling was not performed. A campaign-wide average of wind directions during periods of concurrent mobile sampling is shown by the triangle marker on each wind rose plot. The dashed lines with triangle markers are linear extrapolations of these campaign-wide average wind directions and are meant to guide the eye towards the regions over which these winds travel.

**A6 OA factor maps**

[Figure]

**Figure S18.** Maps of COA and HOA concentrations.

[Figure]

**Figure S19.** Building height data for downtown polygon, obtained from City of Oakland (City of Oakland, 2017). The dashed black border shows limits of the downtown domain used in this study.

[Figure]

**Figure S20.** Black carbon and CO maps.

[Figure]

**Figure S21.** Polygon-specific cumulative distribution function (CDF) curves of black carbon (BC) concentrations. *Insets:* Probability distribution function (PDF) histograms for central tendency statistics (mean and median) of synthetic datasets created using bootstrap resampling of raw data. The PDF histograms are shown with a coarser bin-width to guide the eye better. The abscissae on the insets have the same units as the parent abscissa, but with a zoomed-in scale.